# Haplotype-specific assembly of shattered chromosomes in esophageal adenocarcinomas

## Graphical abstract

## Authors

Jannat Ijaz, Edward Harry,
Keiran Raine, ..., Mathew J. Garnett,
Zemin Ning, Peter J. Campbell

## Correspondence

jijaz@altoslabs.com (J.I.),
pc8@sanger.ac.uk (P.J.C.)

## In brief

Ijaz et al. reconstructed chromothriptic and wild-type chromosomes in patient-derived esophageal adenocarcinoma organoids using a haplotype-aware method. Upon these assemblies, they layered epigenetic and transcriptomic data and found widespread differences between the chromothriptic and wild-type chromosomes within an organoid in topologically associated domains, chromatin accessibility, histone modifications, and gene expression.

## Highlights

- Complex structural variation means haplotype-aware *de novo* assembly methods are essential

- Structural variants change TADs, active and repressed regions, and expression

- Chromothripsis alters epigenome on site-by-site basis rather than global reconfiguration

- Structural variants alter gene expression via ordered change in multiple epigenetic marks

 Ijaz et al., 2024, Cell Genomics 4, 100484
February 14, 2024 © 2023 The Author(s).

# Cell Genomics

CellPress

## Resource

# Haplotype-specific assembly of shattered chromosomes in esophageal adenocarcinomas

Jannat Ijaz,[1,6,*] Edward Harry,[1] Keiran Raine,[1,2] Andrew Menzies,[1] Kathryn Beal,[1] Michael A. Quail,[1] Sonia Zumalave,[3] Hyunchul Jung,[1] Tim H.H. Coorens,[1,4] Andrew R.J. Lawson,[1] Daniel Leongamornlert,[1] Hayley E. Francies,[1,5] Mathew J. Garnett,[1] Zemin Ning,[1] and Peter J. Campbell[1,7,*]

[1]Wellcome Sanger Institute, Hinxton CB10 1SA, UK
[2]Health Innovation East, Unit C, Magog Court, Shelford Bottom, Cambridge CB22 3AD, UK
[3]Mobile Genomes and Disease, Center for Research in Molecular Medicine and Chronic Diseases (CiMUS), Universidade de Santiago de Compostela, 15706 Santiago de Compostela, Spain
[4]Broad Institute of MIT and Harvard, Cambridge, MA, USA
[5]GSK, Gunnels Wood Road, Stevenage SG1 2NY, UK
[6]Present address: Altos Labs, Granta Park, The Portway Building, Great Abington, Cambridge CB21 6GP, UK
[7]Lead contact
*Correspondence: jijaz@altoslabs.com (J.I.), pc8@sanger.ac.uk (P.J.C.)

## SUMMARY

The epigenetic landscape of cancer is regulated by many factors, but primarily it derives from the underlying genome sequence. Chromothripsis is a catastrophic localized genome shattering event that drives, and often initiates, cancer evolution. We characterized five esophageal adenocarcinoma organoids with chromothripsis using long-read sequencing and transcriptome and epigenome profiling. Complex structural variation and subclonal variants meant that haplotype-aware *de novo* methods were required to generate contiguous cancer genome assemblies. Chromosomes were assembled separately and scaffolded using haplotype-resolved Hi-C reads, producing accurate assemblies even with up to 900 structural rearrangements. There were widespread differences between the chromothriptic and wild-type copies of chromosomes in topologically associated domains, chromatin accessibility, histone modifications, and gene expression. Differential epigenome peaks were most enriched within 10 kb of chromothriptic structural variants. Alterations in transcriptome and higher-order chromosome organization frequently occurred near differential epigenetic marks. Overall, chromothripsis reshapes gene regulation, causing coordinated changes in epigenetic landscape, transcription, and chromosome conformation.

## INTRODUCTION

Chromothripsis is a catastrophic genome-shattering event in which a localized region of the genome is fragmented into 10s–100s of pieces and re-ligated in a seemingly random order and orientation.[1] The characteristic genomic patterns of chromothripsis are seen in 29% of cancer genomes, with 3.2% of all chromosomes in cancer cells affected.[2] In several tumor types, chromothripsis is a very early, even initiating, event,[3–6] and it has also been observed in non-cancerous somatic cells.[7,8] Chromothripsis has been observed in the germline, associated with developmental disorders in offspring.[9] Mechanistically, chromothripsis arises as a mitotic catastrophe; lagging chromosomes, dicentric or acentric chromosomes, and sister chromatid fusions can all disrupt anaphase, leading to affected genomic regions being isolated in micronuclei where they undergo extensive fragmentation and rearrangement.[10–15]

In interphase, chromosomes show a high degree of 3-dimensional organization, with specific genomic regions occupying defined locations within the nucleus, maintaining consistently active or repressed chromatin, and showing recurrent patterns of contacts with other genomic loci, both near and distant. Of the many rearrangements in a given chromothripsis event, only one or a few will provide a proliferative advantage to that clone, promoting its progression toward neoplastic growth. However, the high density of rearrangements caused by chromothripsis, regardless of whether they are driver or passenger mutations, is likely to substantially disrupt high-level chromosome organization.

Epigenetic regulation operates over a range of genomic scales, from the nuclear positioning of chromosomes into chromosome domains,[16] through local regulation by enhancer-promoter juxtaposition,[17] to megabase-scale chromosome looping determined by topological-associating domains (TADs).[18] In cancers, TADs are frequently disrupted by

chromothripsis, leading to changes in downstream gene expression patterns.[19] The underlying DNA sequence informs the epigenetic landscape,[20] probably at all scales, but studies relating primary sequence to chromosome organization are hampered by the strong within-species similarity of genome sequence and the high between-species variability of chromosome configuration.

Haplotype-resolved genome assemblies can be used to query allele-specific differences in phenotype, and this is particularly useful when there is a high degree of variation between the two alleles. Since chromothripsis typically affects only one of the two parental chromosomes, cancer clones will often contain a heavily rearranged chromosome (which we will call the "derivative" or "chromothriptic" chromosome) and a chromosome that is considerably less rearranged (which we will call the "wild-type" or "control" chromosome). This property enables assessment of the consequences of chromothripsis on chromatin organization, with the wild-type chromosome representing an internal control for the myriad of factors shaping the epigenetic landscape in that specific cancer clone. Here, we used a suite of long-range DNA sequencing technologies and genome assembly tools to generate high-resolution reconstructions of chromothriptic chromosomes from five esophageal adenocarcinoma organoids, correlating these assemblies with assays of the transcriptional and epigenetic landscape.

## RESULTS

### Genomic and epigenetic characterization of chromothripsis

We studied five esophageal adenocarcinoma organoids in which chromothripsis was fully clonal and affected single parental copies of the chromosome, while the other parental chromosome was retained and largely unrearranged. Two of these organoids were derived from the same patient, one before chemotherapy and the other after relapse (HCM-SANG-0311-C15-B and HCM-SANG-0311-C15, respectively). The patterns of copy number changes and rearrangements of the affected chromosomes satisfied published criteria for classic chromothripsis,[21] with copy number oscillating over two or three states, clustered rearrangements in all four orientations, and alternating regions of retained and lost heterozygosity (Figures 1A–1F and S1). These organoids had driver mutations characteristic of esophageal adenocarcinomas (Figure S2). To generate high-quality haplotype-resolved assemblies, we performed karyotyping (Figures S3A–S3E) and short-read and long-read sequencing on a variety of platforms, including Illumina short-read sequencing, 1M and 8M PacBio continuous long-read (CLR) sequencing, PacBio Circular Consensus Sequencing (CCS) sequencing, and Hi-C chromosome capture. To characterize the transcriptional and epigenetic landscape, we performed Hi-C chromosome capture, long-read RNA sequencing, chromatin accessibility assays, and chromatin immunoprecipitation sequencing (ChIP-seq) for H3K27ac, H3K4me3, and H3K27me3 histone modifications and CTCF binding (Figure 1G and Table S1).

### Genome assemblies

Both reference-based assemblies and *de novo* genome assemblies were used to reconstruct chromothriptic chromosomes, initially using chromosome 6 in HCM-SANG-0300-C15 (Figure 1A) as a pilot. Reference-based genome assemblies, using Gap5,[22] performed poorly in reconstructing chromothriptic chromosomes because of two modes of error: first, contigs would fail to be joined despite plentiful sequencing reads across the breakpoint junction because the contigs were too far apart on the reference genome, and second, contigs would be incorrectly joined together based on reference position rather than the true underlying genomic configuration of the sample.

Haplotype-unaware *de novo* genome assemblies using wtdbg2[23] were also suboptimal for reconstructing the derivative chromosomes. While the control and derivative chromosomes shared many stretches of highly similar sequence, which could be assembled, these segments were punctuated by high divergence between the two chromosomes around rearrangement breakpoints. Current genome assembly methods expect small changes between the two alleles,[24] such as single nucleotide polymorphisms (SNPs), sequencing errors or insertions, and deletions up to kilobases in size. They do not explicitly account for large structural changes within a diploid sample, particularly at the density seen in chromothriptic chromosomes. Since these regions occur so frequently, the resulting assembly is highly fragmented (Table S2).

We found that haplotype-aware *de novo* assemblies overcame many of these problems (Figure 2A). We used WhatsHap[25] to assign germline heterozygous SNPs to phase blocks using PacBio CCS reads. Each block (and the reads assigned to it) was then assigned to either the derivative or the control chromosome using variant allele fractions (VAFs) of SNPs, the presence of chromothripsis-associated structural variants (SVs) and regions of loss of heterozygosity (LOH). Essentially, haplotype blocks generated by WhatsHap that spanned breakpoints of complex rearrangements were assigned to the derivative chromosome, whereas blocks that contained SNPs from regions of LOH were assigned to the wild-type chromosome. For occasional haplotype blocks, typically containing only one or a few germline SNPs that retained heterozygosity in the cancer, we could not resolve which block arose from the derivative and which from the control; these were therefore randomly assigned (but would not be expected to meaningfully affect the assembly beyond the SNPs as they did not span rearrangements).

Having assigned the CCS reads to either the derivative or control chromosome, we then generated an initial, haplotype-aware assembly of each using hifiasm.[26] These initial assemblies were then scaffolded by 3D-DNA[27] using haplotype-resolved Hi-C reads. Haplotype resolution of Hi-C reads was done in a reference-aware manner based on the initial assemblies, using mainly SNPs and differences in mapping quality scores. As scaffolding with 3D-DNA can introduce excessive fragmentation in the initial assembly, this was corrected using CAUS (https://github.com/wtsi-hpag/caus). This method was also used for normal chromosomes and chromosomes with simple rearrangements. In one case when the hifiasm assembly did not accurately reconstruct the underlying sequence, wtdbg2[23] was used (Figure S4).

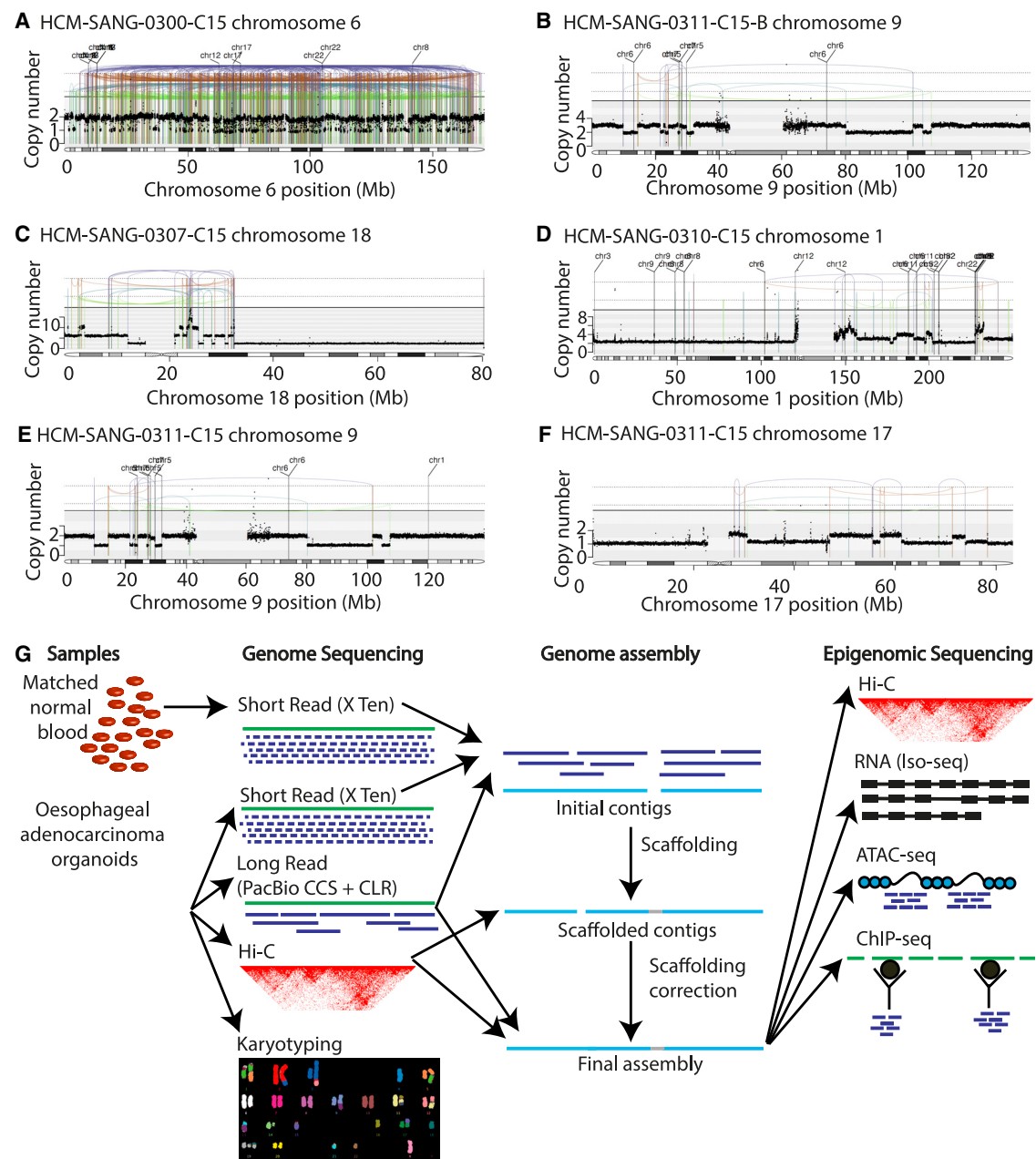

**Figure 1. Chromothripisis present and sequencing modalities used in this study**

(A–F) Rearrangement plots exhibiting chromothriptic regions in each sample. Black dots denote copy number. Vertical lines denote breakpoints: black represents translocations, green represents deletion orientation, teal represents tandem-duplication orientation, dark blue represents tail-to-tail inversion orientation, and brown represents head-to-head inversion orientation. HCM-SANG-0311-C15 has two chromothriptic chromosomes, and all other samples have one chromothriptic chromosome.

(G) Graphical overview of study design. Patient-derived esophageal adenocarcinomas and matched normal blood were used to generate data. Genomic sequencing was used to generate assemblies upon which epigenetic datasets were layered.

The assemblies produced for non-chromothriptic chromosomes were highly contiguous, measured by the smallest number of contigs that constitute 90% of the total assembly size (L90) and the sequence length of the shortest contig at which 90% of the total assembly size is reached (N90). The median L90 was 2 Mb (Figure S5A), and the median N90 was 18.3 Mb (Figure S5B).

Our method produces assemblies that represent the maternal and paternal haplotypes in the dominant clone of the organoid. However, most cancer samples, including those sequenced

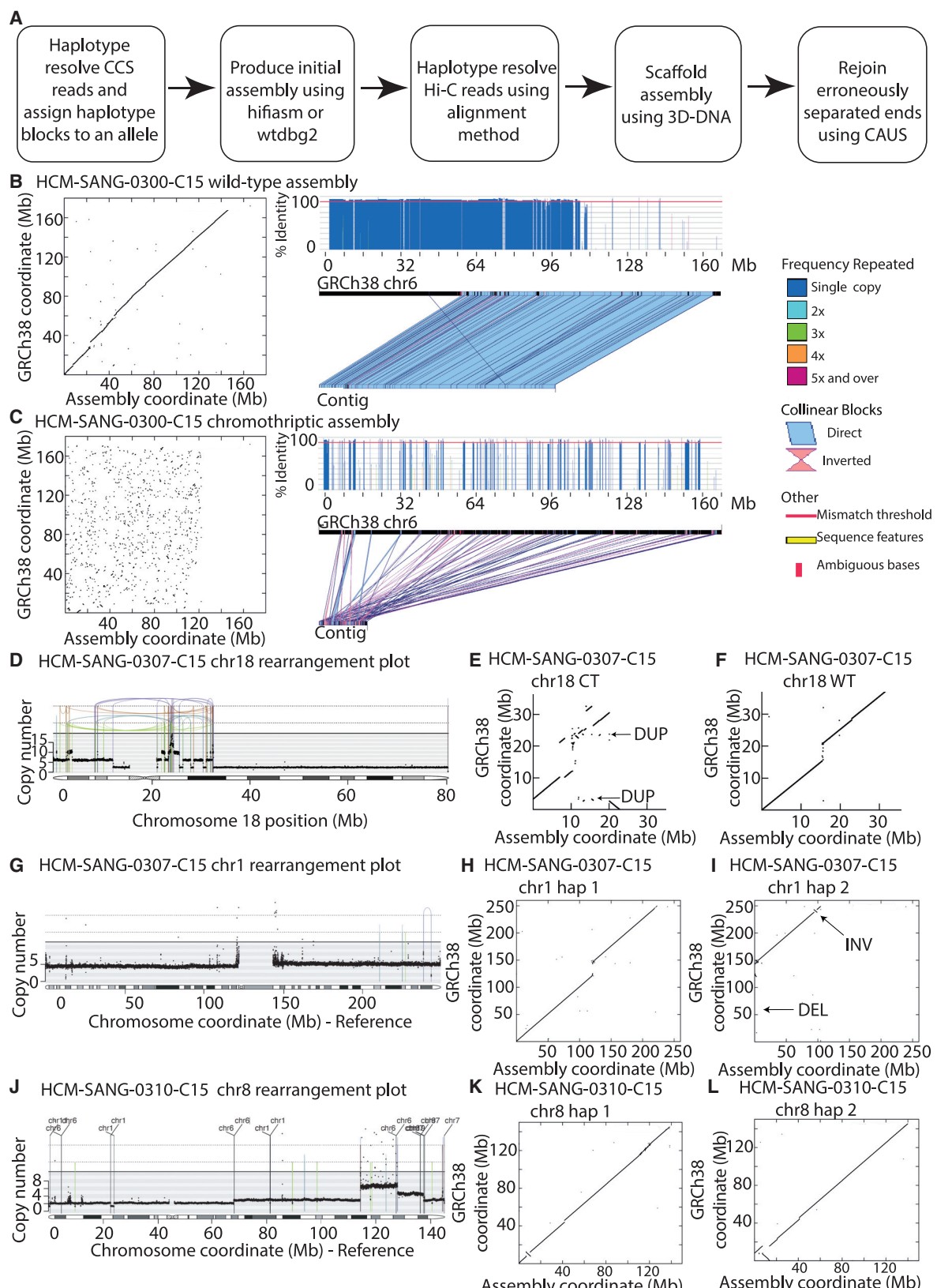

**A**

Haplotype resolve CCS reads and assign haplotype blocks to an allele → Produce initial assembly using hifiasm or wtdbg2 → Haplotype resolve Hi-C reads using alignment method → Scaffold assembly using 3D-DNA → Rejoin erroneously separated ends using CAUS

**B** HCM-SANG-0300-C15 wild-type assembly

**C** HCM-SANG-0300-C15 chromothriptic assembly

Frequency Repeated
- Single copy
- 2x
- 3x
- 4x
- 5x and over

Collinear Blocks
- Direct
- Inverted

Other
- Mismatch threshold
- Sequence features
- Ambiguous bases

**D** HCM-SANG-0307-C15 chr18 rearrangement plot

**E** HCM-SANG-0307-C15 chr18 CT

**F** HCM-SANG-0307-C15 chr18 WT

**G** HCM-SANG-0307-C15 chr1 rearrangement plot

**H** HCM-SANG-0307-C15 chr1 hap 1

**I** HCM-SANG-0307-C15 chr1 hap 2

**J** HCM-SANG-0310-C15 chr8 rearrangement plot

**K** HCM-SANG-0310-C15 chr8 hap 1

**L** HCM-SANG-0310-C15 chr8 hap 2

*(legend on next page)*

here, will contain aneuploidy (Figure S3). Our method was most effective when the two alleles had an asymmetric copy number, such as two copies of one parental chromosome and one copy of the other resulting from whole chromosome duplications affecting only one haplotype. In these cases, read depth becomes informative for read assignment. The effect of asymmetric copy number is evident when comparing chromosome 9 in the initial and relapse samples, where the initial sample has two copies of the wild-type chromosome (Figures S5C–S5E), and the relapse has one copy of the wild-type chromosome (Figures S5F–S5H). Both contained a single copy of the chromothriptic chromosome. While the resultant assemblies are very similar, the asymmetric copy number and associated increased read depth leads to a more contiguous assembly of the wild-type chromosome (asymmetric copy number N90 = 40,086,745 and L90 = 2 and symmetric copy number N90 = 5,427,947 and L90 = 5). The chromothriptic assemblies were more similar (N90 of 3,358,574 and L90 of 5 in the initial samples and N90 of 5,819,878 and L90 of 4 in the relapse sample).

### Reconstruction of rearranged chromosomes

This haplotype-oriented *de novo* assembly method led to contiguous cancer genome assemblies, even in highly rearranged chromosomes (Tables S3 and S4). For example, chromosome 6 from HCM-SANG-0300-C15 with over 900 rearrangements reconstructed into 50 contigs with an L90 of 11 (Table S3). Reassuringly, the assembled contigs for the control chromosome were structurally nearly identical to the reference genome, allowing for small germline insertions and deletions, even though the reference genome was not used at all to guide the assembly (Figure 2B). Conversely, the contigs from the derivative, chromothriptic chromosome consisted of haphazardly organized segments originating from throughout the reference chromosome, consistent with the breakage and repair caused by the chromothripsis (Figure 2C).

The assemblies of the chromothriptic chromosomes in the other samples exhibited similar patterns (Figures 2D–2F and S5C–S5N). Together, these findings suggest that haplotype-specific resolution and subsequent genome assembly produces contigs that are highly representative of the underlying sequence and are therefore suitable for identifying differences between the derivative and control chromosome in transcriptional and epigenetic phenotypes of interest.

### Reconstructing other classes of SV

Chromothripsis produces extreme divergence between the two parental copies of a given chromosome, which may make haplotype assignment and assembly more accurate. We therefore assessed whether our assembly method was able to reconstruct other classes of structural variation. Deletions were the simplest, most accurate type of SV to reconstruct because true clonal deletions have no reads in the deleted region and reads spanning the deletion junction (Figures 2G–2I). Reciprocal inversions can be difficult to detect in short-read sequencing because they are copy number neutral, but the long-read assemblies generated successful reconstructions (Figures 2G–2I).

Our method struggled to correctly reconstruct duplications once they were longer than the average read length. In cases where duplications were small enough to have single reads that crossed both sides of the duplicated segment, our method could incorporate multiple copies of the segment in the final assembly, such as seen in the chromothriptic chromosome 18 in HCM-SANG-0307-C15 (Figures 2D–2F). However, with long duplicated segments, we found that they were typically collapsed into a single copy in the assembly, as exemplified by chromosome 8 in HCM-SANG-0310-C15 (Figures 2J–2L). This inability to reconstruct tandem duplications is not caused by incorrect haplotype resolution, as large duplications in a haploid chromosome were also not reconstructed (Figures S5O–S5Q). Rather, it arises because somatic duplications have high sequence similarity, and current assembly methods typically do not incorporate read depth or copy number data.

Subclonal variants are common in cancer genomes, and these are difficult to assemble. This is because the read coverage in regions of subclonality is not equal to regions that are clonal. As such, most commonly these regions get collapsed by the assembler to resemble the dominant clone. Subclonal deletions with a high VAF often do not have enough read coverage to assemble the region leading to a fragmented assembly (Figures S6A–S6D). In general, as the number of cells containing the subclonal deletion decreases, the likelihood of generating that region increases as there are more reads that will support assembling the region. Similar to clonal duplications, subclonal duplications are often collapsed by assemblers as most do not account for read depth (Figures S6E–S6H).

Extrachromosomal DNA can also be difficult to assemble. These small, highly amplified regions are difficult to assemble because the short fragments cannot be joined to other sequences of DNA. In short-read sequencing, such regions are identifiable as regions of massive copy number gains with very sharp boundaries[28,29]; we did not see these patterns in the organoids analyzed here, suggesting that extrachromosomal DNA was largely absent from our samples.

Haplotype-aware assembly could also reconstruct other types of complex structural rearrangement, including breakage-fusion-bridge cycles and regions where there had been fusion and subsequent rearrangement of two chromosomes (Figures S7A–S7N). However, as for simple tandem duplications,

---

**Figure 2. Genome assemblies of complex rearrangements**
(A) Main steps in method for cancer genome assembly.
(B and C) Left: dot plot alignments of the wild-type and chromothriptic assemblies, respectively, against the reference GRCh38 chromosome 6. A sequence identical to the reference GRCh38 genome would appear as a 45° diagonal line. Right: alignment plots produced using XMatchView for largest contigs from the wild-type and chromothriptic assemblies, respectively. Black line represents chromosome 6 from GRCh38, and the line below represents the contig. Blue regions are direct repeats, and pink regions are inverted repeats. An identity threshold was set at 90%.
(D, G, and J) Rearrangements plots are as previously described.
(E, F, H, I, K, and L) Dot plots alignments of each haplotype to the reference GRCh38 genome. Hap, haplotype; CT, chromothripsis; WT, wild-type.

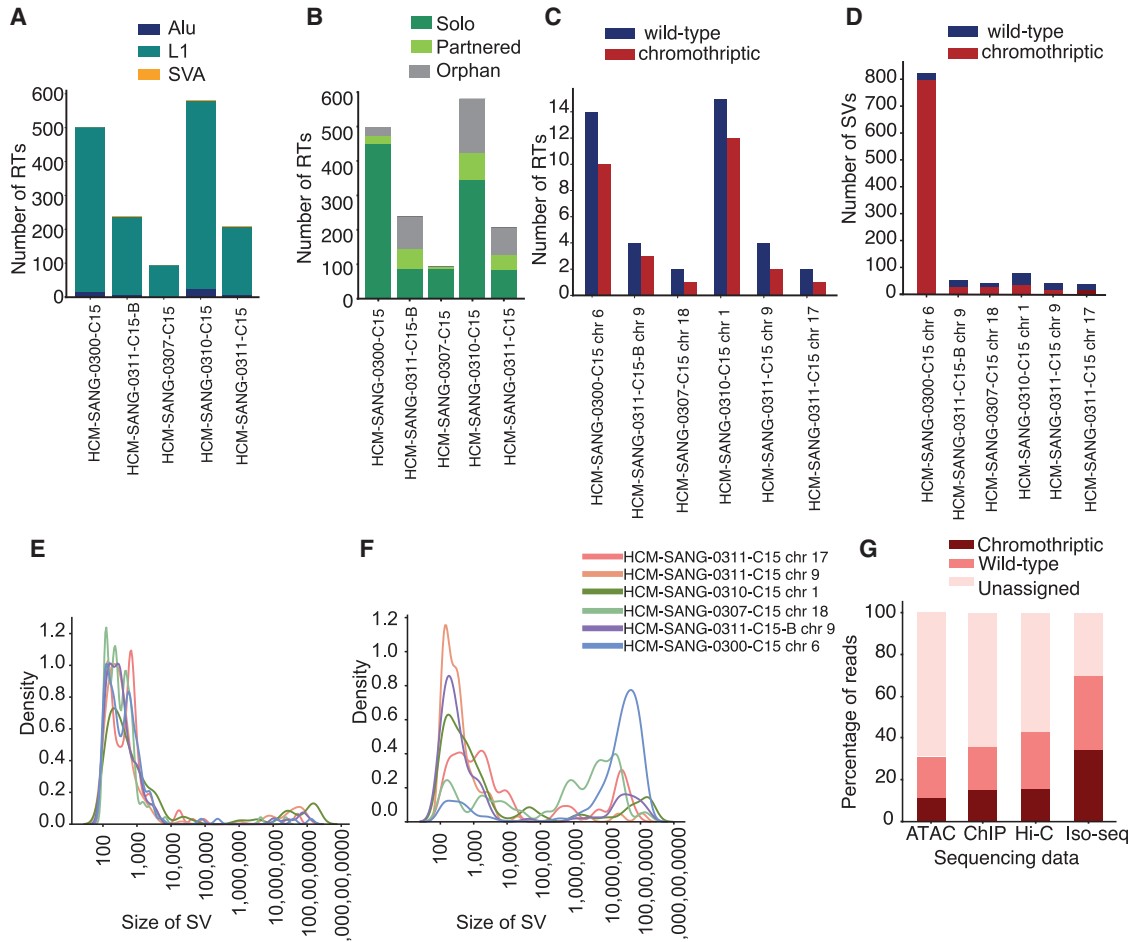

**Figure 3. Haplotype-resolved structural rearrangements, retrotransposons, and epigenetic modalities**
(A) Total retrotransposon classes in each sample. L1, LINE-1; SVAs, SINE-VNTR-Alus.
(B) Total retrotransposon types in each sample.
(C) On the chromosomes with evidence of chromothripsis, number of retrotransposon calls.
(D) Total SV counts greater than 1 kb, filtered to remove germline SVs and SVs in repeat regions, which are likely to be erroneous calls.
(E and F) SV size distribution for wild-type and chromothriptic chromosomes, respectively.
(G) Proportion of reads assigned in different data types.

large duplications in these regions of complex rearrangement were often not successfully reconstructed.

**Haplotype-resolved SV calls**
With these haplotype-resolved genome assemblies, allele-specific structural variation was assessed in the long reads, including both somatic retrotransposon insertions and SVs. LINE-1 elements were the most common somatic retrotransposon event in all samples, while SINE-VNTR-Alus (SVAs) and *Alu* retrotranspositions represented minor categories (Figure 3A). Most of the observed retrotranspositions involved solo insertions of LINE-1 elements. However, there was also evidence indicating the occurrence of partnered and orphan transductions involving distinct downstream sequences. Similar findings have been described in short reads on cancer genomes[30,31] (Figure 3B). A similar number of events were seen on wild-type and chromothriptic chromosomes (Figure 3C), suggesting that

chromothripsis did not hugely perturb the rate of retrotransposition. The majority of somatic retrotranspositions were identified with both long-read and short-read sequencing, with more overall from long reads (Figure S8A). The long-read-specific events were more frequently found in regions with very low GC content (Figure S8B) and frequent repeats (Figure S8C), suggesting the long-read sequencing is more accurate in areas that are difficult to map with short reads.

As expected, the patterns of SVs differed between chromothriptic and wild-type chromosomes (Figure 3D). For example, in chromosome 6 in HCM-SANG-0300-C15, we called 799 somatic SVs greater than 1 kb on the derivative chromosome and 24 somatic SVs on the control chromosome. The size distribution of SVs on the chromothriptic and wild-type chromosomes differed. Focusing on chromosome 6 in HCM-SANG-0300-C15, most SVs (81%) on the control chromosome were under 1,000 bp in size (Figure 3E). In contrast, the derivative

chromosome showed a bimodal distribution, with only 12% SVs under 1,000 bp, although this was comparable in absolute numbers to the number of small SVs on the wild-type chromosome. For 84% of SVs on the derivative chromosome, the two break-ends joined in an SV were separated by more than 1 Mb on the reference genome (Figure 3F).

### Haplotype resolution of epigenetic and transcriptomic datasets

Organoids were profiled using ChIP-seq for two active histone marks, H3K27ac and H3K4me3, and one repressive histone mark, H3K27me3. They were also profiled for CTCF, a TAD boundary marker. Overall chromatin accessibility was studied using the assay for transposase-accessible chromatin with sequencing (ATAC-seq).[32] Transcriptomes were profiled using long-read RNA sequencing (Iso-seq).[33] Using the haplotype-resolved genome assemblies, we assigned reads from the epigenomic and transcriptomic datasets to the derivative and control chromosomes using SVs and germline heterozygous SNPs reported in the sequencing reads. The proportion of reads that can be haplotype resolved in this way was heavily dependent on the underlying read length, with longer reads more likely to traverse a heterozygous SNP and therefore be assigned. Iso-seq reads had the highest proportion of assigned reads, and ATAC-seq reads had the lowest (Figure 3G). We then used DiffBind[34] to identify chromatin peaks that had differential heights between the two parental haplotypes, DESeq2[35] to identify differentially expressed genes, and Cooltools[36] to identify TAD boundaries.

### Allele-specific chromatin accessibility and histone modifications

The motivation for haplotype-resolving epigenomic datasets is to assess whether wild-type and chromothriptic chromosomes have materially different profiles. Of course, two homologous chromosomes may differ for reasons not related to chromothripsis such as germline variants, point mutations, and so on. In organoids without chromothripsis on chromosome 6 (Figures S8D, and S8E), for example, we found between 7% and 11% of peaks showed haplotype-specific differences in peak heights on this chromosome. Furthermore, these peaks were stable over time. Comparing the two samples collected from the same donor, one at first diagnosis (HCM-SANG-0311-C15-B) and one post relapse (HCM-SANG-0311-C15), 610 peaks were present in both samples in regions we could confidently haplotype resolve. We found similar peak heights in both samples (Pearson correlation coefficient r = 0.69, p = $1.74 \times 10^{-176}$; Figure 4A), and when comparing individual histone modifications and CTCF binding sites, we found a large overlap in the marks that were found on each haplotype across samples (Figure S9). This suggests that histone modifications and chromatin accessibility are stable. Even chromosome 9, for which one parental copy is chromothriptic in both samples, showed stable patterns of chromatin peaks over time (Pearson correlation coefficient r = 0.80, p = $1.3 \times 10^{-234}$; Figure 4B) with a large overlap of individual histone modifications and CTCF binding sites (Figure S9). The baseline expectation, then, under our assay conditions on these tumor organoids, is that approximately 10% of ChIP-seq and ATAC-

seq peaks show differential binding between the two parental haplotypes.

Turning to chromosome 6 in HCM-SANG-0300-C15, where chromothripsis has generated ~900 SVs, we could haplotype resolve 2,530 peaks across the ChIP-seq and ATAC-seq data. Of these, 1,337 (52.8%) peaks were differential between the two parental chromosomes (Figure 4C, Wald test, q < 0.05). This proportion is significantly greater than that for chromosome 6 in organoids without chromothripsis of this chromosome (HCM-SANG-0300-C15 versus HCM-SANG-0311-C15 odds ratio 5.2; 95% confidence interval 4.8–6.9; p < $10^{-16}$; Fisher's exact test; comparisons with other samples were similar). We examined potential mechanisms causing differential binding on chromosome 6 of HCM-SANG-0300-C15. Simple dosage effects, such as duplication (Figure S10) or LOH on one allele (Figure S11), caused the differential activity of 1,034 (40.9% of total) peaks. For a further 39 (1.5% of total), differential peaks occurred because a chromothriptic SV directly intersected and disrupted a peak (Figure S12).

This left 264 (10.4% of total) peaks that did not have simple explanations for their differential activity; this fraction was similar to the numbers of differential peaks on chromosomes without chromothripsis. Nonetheless, compared to the set of peaks that were equally active on the two chromosomes, these remaining differential peaks were significantly closer to SVs on the chromothriptic chromosome. Weaker peaks had a median distance of 45 kb from nearest SV, and stronger peaks had a median distance of 56 kb, whereas non-differential peaks had a median distance of 67 kb (p = $5.6 \times 10^{-5}$ and p = 0.012 for peaks with lower or greater activity on the chromothriptic chromosome, respectively; Wilcoxon rank-sum test; Figures 4D and 4E). This was not the case for distance to the nearest SV on the wild-type chromosome, implying that it is the rearranged genomic architecture generated by chromothripsis that has led to the differential peaks (median distance to SV 548 kb for stronger peaks, 551 kb for weaker peaks, and 614 kb for non-differential peaks; p = 0.96 and p = 0.39, respectively; Wilcoxon rank-sum test; Figures 4D and 4E). The density distributions showed a shoulder of differential peaks within 10 kb of an SV on the chromothriptic chromosome that was not evident for non-differential peaks. 18% of peaks that are weaker on the chromothriptic and 16% of peaks that are stronger on the chromothriptic chromosome are less than 10 kb from a SV compared to 9% of non-differential peaks being less than 10 kb. In contrast, on the wild-type chromosome, 0%, 3.2%, and 3.3% of peaks that are stronger on the wild-type, weaker on the wild-type, and non-differential were less than 10 kb from an SV. This effect applied across different histone peaks assessed as well as CTCF binding sites (Figures 4F and 4G).

Taken together, these data suggest that chromothripsis can extensively reorganize a chromosome's epigenetic landscape. In our most severely rearranged organoid, just over half of the peaks were differentially active between the chromothriptic and wild-type chromosomes; while ~80% of these differential peaks could be explained by dosage effects or direct disruption by SVs, the remaining 20% arose from less direct alterations, often acting over the scale of several kilobases. Similar effects were seen for chromothriptic chromosomes in the other samples (Figure S13A and S13B), suggesting that this is a generalizable feature of chromothripsis.

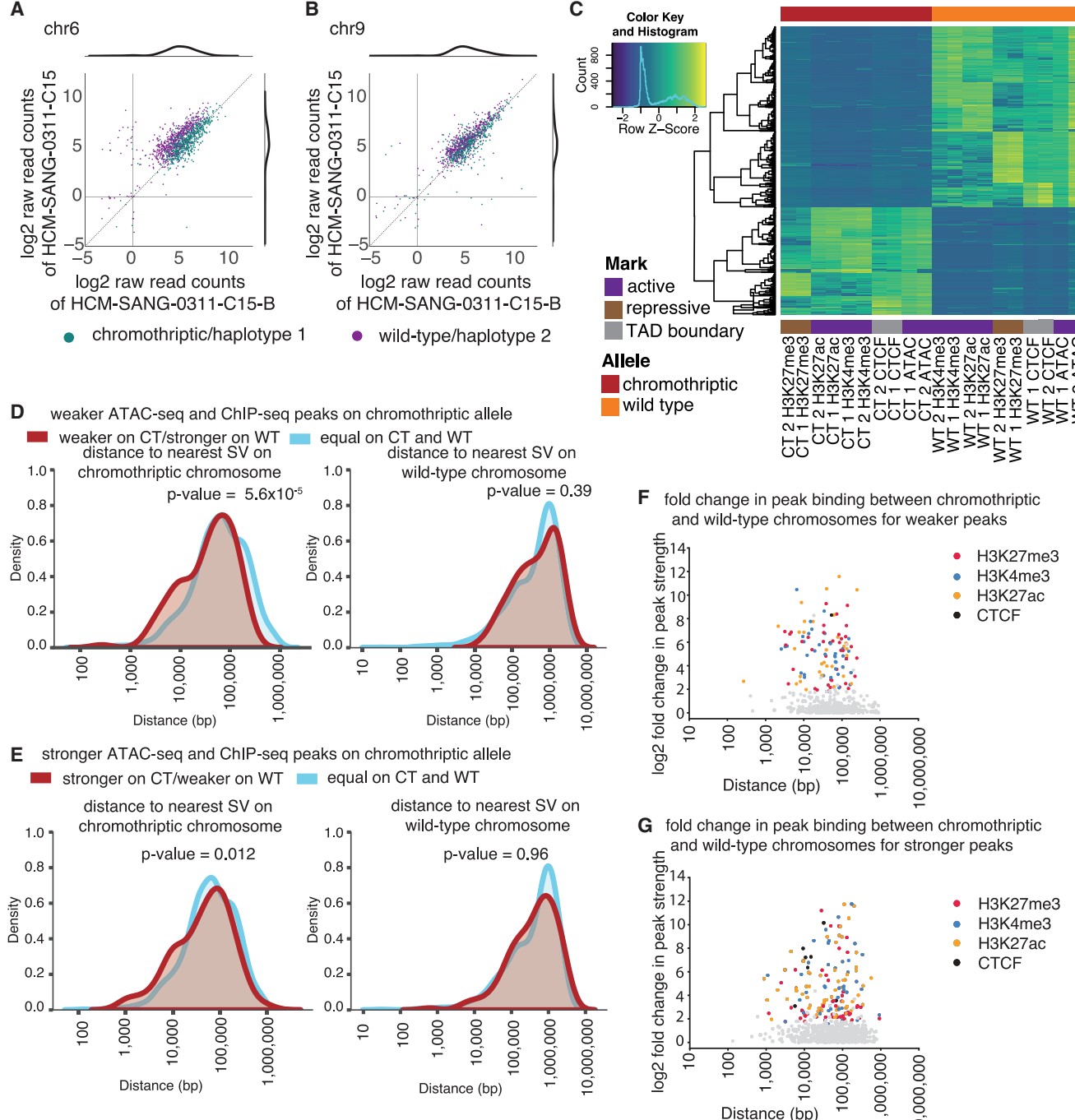

**Figure 4. Differences in chromatin accessibility and histone modification between alleles**

(A and B) Log2 raw read counts of peaks that can be assigned in both samples. Marginal density plots are shown.

(C) Heatmap of differentially bound and open chromatin sites on the wild-type and chromothriptic chromosome 6 in HCM-SANG-0300-C15.

(D) Distance of peaks that are weaker on the chromothriptic chromosome to the nearest SV on the chromothriptic chromosome (left) and wild-type chromosome (right) relative to non-differential peaks. p values were calculated using the Wilcoxon rank-sum test.

(E) Distance of peaks that are stronger on the chromothriptic chromosome to the nearest SV on the chromothriptic chromosome (left) and wild-type chromosome (right) relative to non-differential peaks. p values were calculated using the Wilcoxon rank-sum test.

(F and G) Distance effect on fold change of peak strength split by mark profiled using ChIP-seq.

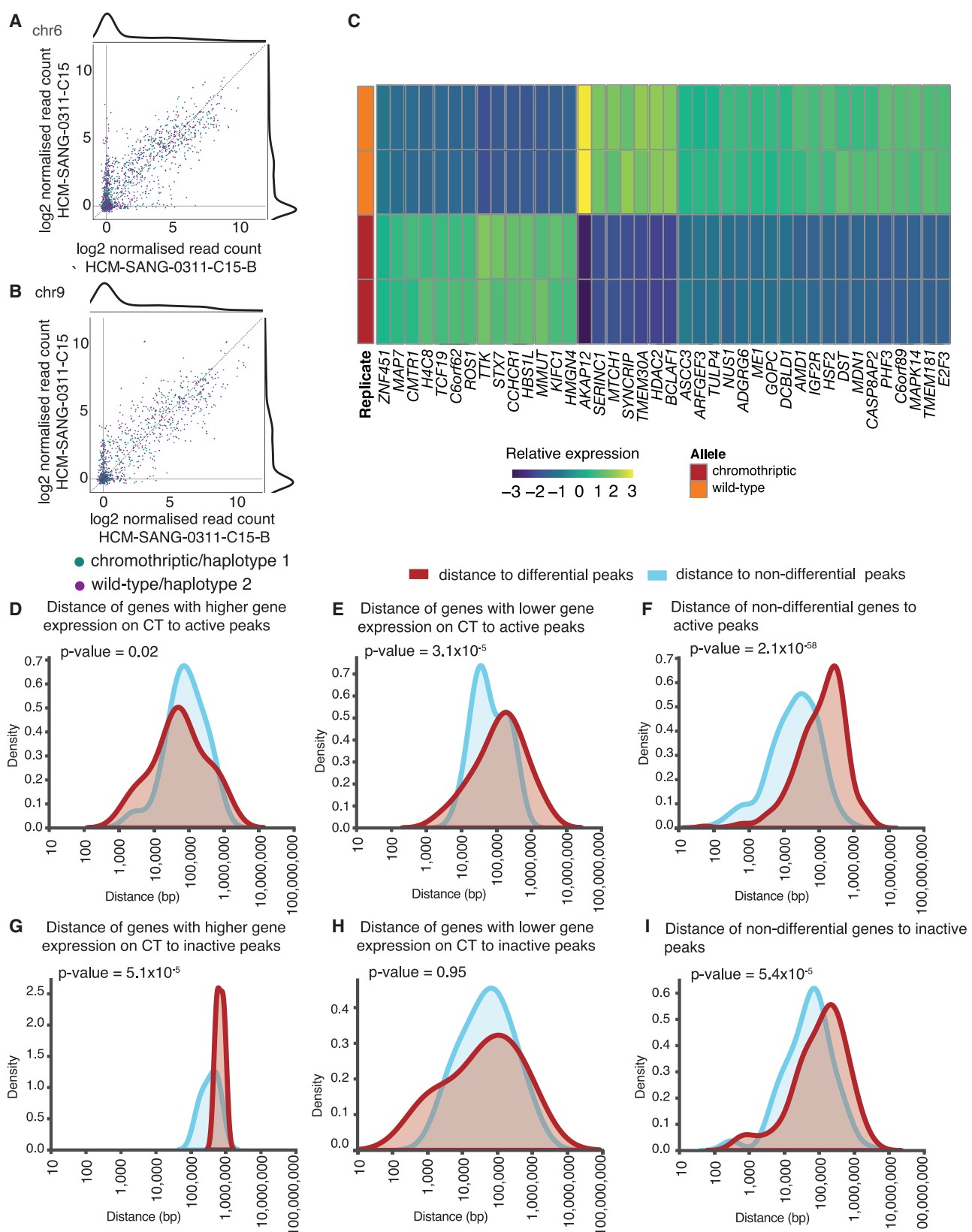

## Allele-specific gene expression

We also examined allele-specific differences in gene expression using the haplotype-resolved genome assemblies. As above, to establish baseline rates of allele-specific differences in gene expression, we studied chromosome 6 in HCM-SANG-0311-C15-B and HCM-SANG-0311-C15, which do not carry chromothripsis on the chromosome. Overall, 2.3% of transcripts for HCM-SANG-0311-C15-B and 1.3% of transcripts for HCM-SANG-0311-C15 showed differential expression between the two alleles. These frequencies are lower than those seen for chromatin peaks, suggesting that chromatin accessibility and histone modifications are more variable between parental haplotypes than gene expression or that the assays that assess these are more variable.

Since HCM-SANG-0311-C15-B and HCM-SANG-0311-C15 are derived from the same donor at different time points, we could also assess the stability of gene expression over time. Genes that are, for example, highly expressed on one haplotype in HCM-SANG-0311-C15-B were typically also highly expressed on the same haplotype in HCM-SANG-0311-C15, suggesting that gene expression is stable over time (1,866 genes, Pearson's correlation coefficient r = 0.83, $p < 1 \times 10^{-16}$) (Figure 5A). Interestingly, there were 224 genes that have no expression on chromosome 6 in the initial sample but become expressed in the relapse sample and 117 that lost expression in the relapse sample. Most of these alterations in gene expression were likely to be passenger events as only one gene (CCND3) has been reported as a cancer driver by the Oesophageal Cancer Cilinal and Molecular Straificaion (OCCAMS) consortium[37] or the Pan-Cancer Analysis of Whole Genomes(PCAWG) study.[38] Similar gene expression stability was observed when comparing chromothriptic chromosome 9 and wild-type chromosome 9 in the two samples (1,176 genes with no change, 133 gaining expression, 68 losing expression between initial and relapse samples; Pearson's correlation coefficient r = 0.86; $p < 1 \times 10^{-16}$; Figure 5B).

Due to the high density of SVs, chromosome 6 in HCM-SANG-0300-C15 was used to study the relationship between primary genome sequence and gene expression. Differential expression between chromothriptic and control chromosomes was determined using DESeq2[35] from all reads confidently assigned to a single haplotype. There were 229 differentially expressed genes (Wald test, q value < 0.05), representing 9.4% of total genes expressed from chromosome 6 in regions that could be confidently haplotype resolved (2,432), a greater number than seen comparing chromosomes without chromothripsis (1.3%–2.3%) (HCM-SANG-0300-C15 versus HCM-SANG-0311-C15 odds ratio 13.5; 95% confidence interval 7.4–27.6; $p < 10^{-16}$; Fisher's exact test; comparisons with other samples were similar). Relative to the non-chromothriptic parental copy, 102 genes had higher expression, and 127 had lower expression on the chromothriptic copy (Figure 5C). Most of the disrupted genes are likely to be passenger mutations. When compared to cancer genes reported by PCAWG[38] and OCCAMS,[37] only MAP3K4 on chromosome 6 was disrupted by chromothripsis in HCM-SANG-0300-C15. Turning to the chromothriptic chromosome in other samples, CDKN2A was deleted by chromothripsis on chromosome 9 in HCM-SANG-0311-C15. There were also very few copy number drivers on chromothriptic chromosomes in the other samples, only 2 on chromosome 1 in HCM-SANG-0310-C15, and 1 on chromosome 9 in HCM-SANG-0311-C15-B.[39]

Of those 229 differentially expressed genes on chromosome 6 in HCM-SANG-0300-C15, copy number alterations overlapped 109 (4.5% of total). For a further 60 genes (2.5% of total), the gene had been fragmented by a chromothriptic SV, which generally prevented complete transcripts being generated. This left 60 (2.5% of total) genes whose differential expression could not be attributed by copy number alterations or simple breakage by SVs. As for ATAC and chromatin peaks, this is a similar percentage of genes as seen on chromosomes without chromothripsis.

The remaining genes that are upregulated and those that are downregulated on the chromothriptic chromosome were not closer to SVs than non-differential peaks (median distance to nearest SV for genes with lower expression 103 kb, genes with higher expression 72 kb versus non-differential genes 80 kb; p = 0.71 and p = 0.22, respectively; Wilcoxon rank-sum test; Figures S14A–S14C). Similar effects were seen for chromothriptic chromosomes in the other samples (Figures S14G and S14H). However, the differentially expressed genes were closer to differential chromatin/ATAC peaks than to non-differential peaks (Figures S14D–S14F). Differential genes that have higher expression on the chromothriptic allele were closer to active peaks than non-differential peaks (median distance of genes to an active mark was 46 kb, and median distance of non-differential genes was 75 kb [n = 20]; p = 0.02; Wilcoxon rank-sum test; Figures 5D–5I). This suggests that the differential gene expression arises in an altered chromatin landscape.

**Figure 5. Differences in gene expression between alleles**

(A and B) Log2 raw expression of genes that can be assigned in both samples. Marginal density plots are shown.

(C) Heatmap of 40 most differentially expressed genes on the wild-type and chromothriptic chromosome 6 in HCM-SANG-0300-C15. There is high concordance between biological repeats.

(D) Distance of differential genes with higher gene expression on the chromothriptic allele to active peaks. The p value was calculated using the Wilcoxon rank-sum test.

(E) Distance of differential genes with lower gene expression on the chromothriptic allele to active peaks. The p value was calculated using the Wilcoxon rank-sum test.

(F) Distance of non-differential genes to active peaks. The p value was calculated using the Wilcoxon rank-sum test.

(G) Distance of differential genes with higher gene expression on the chromothriptic allele to inactive peaks. The p value was calculated using the Wilcoxon rank-sum test.

(H) Distance of differential genes with lower gene expression on the chromothriptic allele to inactive peaks. The p value was calculated using the Wilcoxon rank-sum test.

(I) Distance of non-differential genes to inactive peaks. The p value was calculated using the Wilcoxon rank-sum test.

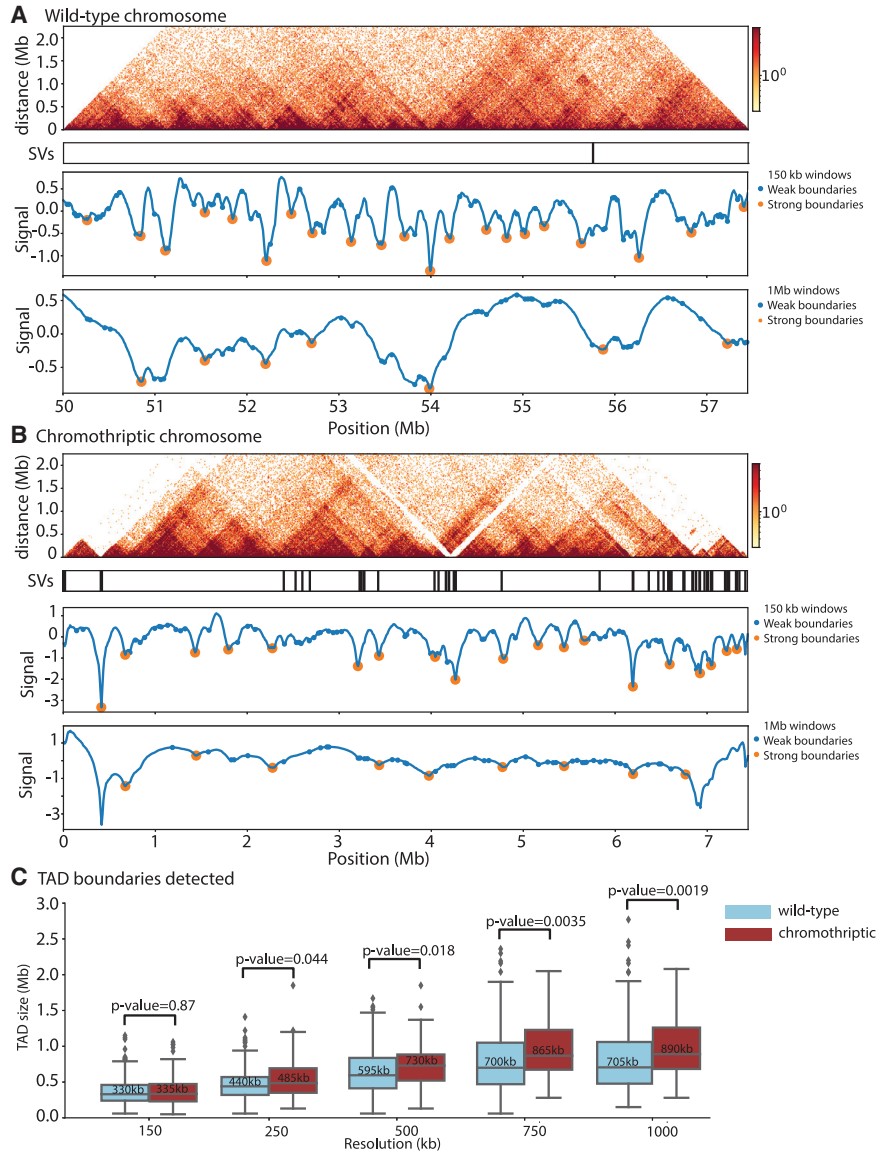

**Figure 6. Differences in higher-order chromatin conformation between alleles**

(A and B) Example TAD boundary calls on wild-type and chromothriptic chromosomes, respectively. Top: Hi-C contact matrix with SV track showing where structural variants are present relative to the reference genome. Middle: TAD boundary calls using 150-kb bins. Signal values represent the difference in number of contacts between adjacent bins. When the differences are considered to represent TAD boundaries, orange dots are plotted. These correspond with small TAD structures. Bottom: TAD boundary calls using 1-Mb bins as described above. These represent larger TAD structures.

(C) TAD sizes on wild-type versus chromothriptic chromosomes in HCM-SANG-0300-C15 called using different bin sizes. TADs are inferred as regions between boundary calls. p values were calculated using the Wilcoxon rank-sum test. Median values are stated on boxplots.

chromosome were, on average, larger than those on the wild-type chromosome (p = 0.044, p = 0.018, p = 0.0035, and p = 0.0019 for 250 kb, 500 kb, 750 kb, and 1 Mb, respectively, Wilcoxon rank-sum test; median distances quoted in Figure 6C). This suggests that after chromothripsis, the larger-scale chromosome organization in this sample was disrupted, with more contacts between distant regions of the genome and less strict compartmentalization of different segments into TADs. On chromothriptic chromosomes on other samples, the TAD sizes on chromothriptic and wild-type chromosomes were very similar at all resolutions, suggesting that a high density of breakpoints is needed to systematically alter TAD sizes (Figure S15).

To assess the interaction between multiple modes of regulation, we layered Hi-C chromosome capture data onto regions with differential epigenetic peaks and allele-biased transcription (Figures 7 and S16–S18). *AKAP12* is the most differentially expressed gene in HCM-SANG-0300-C15 (540-fold higher on WT chromosome; q = 2.3 × 10$^{-117}$, Wald test); it retains its usual reference configuration in the wild-type chromosome, but the complete gene footprint has been sandwiched between usually distant segments of chromosome 6 on the chromothriptic copy (Figure 7). On the wild-type chromosome, H3K27ac and H3K4me3 peaks are present at the start of the *AKAP12* gene, suggestive of an active chromatin landscape, but these peaks are significantly reduced on the chromothriptic chromosome. On the wild-type chromosome, there are chromatin contacts between the start of the *AKAP12* gene and an upstream region (Figure 7; Hi-C track, arrow A), which is a likely enhancer given that it is also marked with H3K27ac (Figure 7, boxes). This enhancer is moved

## Topological associated domains

The changes in chromatin and transcriptional landscape described above operate in a 3-dimensional organization of the rearranged chromosome. To quantify this, we measured changes in contact frequencies on both the chromothriptic and wild-type chromosome in HCM-SANG-0300-C15 in order to call TAD boundaries at a variety of resolutions (Figures 6A and 6B). The distance between boundaries can then be used to infer TAD size. Large-scale rearrangement of primary genome sequence is likely to disrupt TAD structures at a variety of levels, so to quantify this, 5 different resolutions were used to allow different TAD hierarchies to be called (Figure 6C). The TAD sizes were similar at a 150-kb resolution (median TAD size 335 kb on chromothriptic versus 330 kb on wild-type chromosome; p = 0.87; Wilcoxon rank-sum test). However, for larger-scale chromosome organization (TAD calling at resolutions of 250 kb or higher), TADs on the chromothriptic

**Cell Genomics**
**Resource**

**Figure 7. A 600-kb region of the custom assemblies**

Wild-type assembly on the left and chromothriptic assembly on the right. The regions shaded in gray are identical sequences in the two chromosomes, if ignoring indels, and SNPs and are the same as the sequence chr6:151,209,484–151,396,541 in the GRCh38 reference genome. This gray shaded region contains AKAP12 and ZBTB2. Colored blocks in the contiguous sequences track are contiguous sequences found in the reference genome. A block is contiguous but is not found in the reference GRCh38 genome adjacent to the next block. The arrows denote orientation of the regions relative to the reference GRCh38 genome. Each pixel in the Hi-C track is an interaction. Differential ChIP-seq and ATAC-seq peaks were called using DiffBind, and the scale measures relative peak score. The peak is shown on the chromosome where that mark is upregulated. There are multiple differentially active sites on the wild-type chromosomes and no differential sites on the chromothriptic chromosomes in this region. Histone tracks (H3KXXX) show raw haplotype-resolved histone read coverages. In Iso-seq tracks, black lines show splicing, and gray boxes represent exons.

chromosome, likely caused by the *FRK* being moved downstream of *PACRG-AS1*. Since *FRK* is transcribed, this leads to both fusion transcripts of FRK joined to *PACRG-AS1* and normal *PACRG-AS1* transcripts (Figure S18).

**DISCUSSION**

Using long-read sequencing technologies, we assembled highly rearranged chromosomes and used these to query allele-specific changes in gene expression, epigenetic changes, and higher-order structuring. Chromosomes with simple rearrangements may be reconstructed by generating a single assembly for both alleles, known as a collapsed assembly, and then separating the two haplotypes. However, chromosomes with extreme structural differences between parental copies cannot be so simply reconstructed. At regions where an identical sequence in the two chromosomes becomes divergent, we found that a haplotype-unaware approach could not stitch together both parental copies, and this led to a very fragmented assembly. Haplotype-aware methods circumvent this problem by segregating reads and reducing, though not totally eliminating, the number of instances in which fragmentation occurs. However, segregation of reads required both long and highly accurate reads. While it is technically possible to reconstruct real-world primary patient samples using our method, it requires high sequencing depth, and only the dominant maternal and paternal clones will be assembled. As such, large biopsies from relatively clonal samples will be required, and

elsewhere by structural variation on the chromothriptic chromosome, and the segments that replace it show no histone peaks and make minimal contact with the *AKAP12* transcription start site; loss of this enhancer presumably explains the decrease in *AKAP12* expression from the chromothriptic chromosome. Interestingly, the adjacent gene, *ZBTB2*, is not differentially expressed between the two copies (q = 0.98, Wald test), and histone peaks at *ZBTB2* are similar on chromothriptic and wild-type chromosomes.

Alteration of other genes on chromosome 6 can also be understood by integrating multiple epigenetic and transcriptomic layers. One example is *TPD52L1* where the histone landscape looks similar between the chromosomes, but chromatin contacts are lost (Figure S16). Several novel fusion genes involving *MYO6* resulted from structural variation creating new chromatin contacts, histone peaks, and abnormal fusion transcripts (Figure S17). *PACRG-AS1* is upregulated on the chromothriptic

care must be taken to avoid contamination with normal tissue. We opted for PacBio CCS to generate the initial assemblies, but the pipeline is also compatible with nanopore long-read sequencing.

Haplotype-specific alterations caused by chromothripsis can be mechanistically important to the biology of cancer. Here, we used esophageal adenocarcinoma organoids with chromothripsis to understand how changes in primary genome structure affect higher-order chromatin biology. The answer is far from simple. We found that the wholesale chromosome shuffling caused by chromothripsis alters patterns of chromatin contacts, histone modifications, chromatin accessibility, and gene expression, with frequent interplay between these different levels of effect.

We presume that the vast majority of the rearrangements are passenger events, not conferring a selective advantage on the cancer clone. However, selection relies on genomic changes that have phenotypic consequences for the cell; our data show that, for chromothripsis, maybe 10%–20% of SVs do alter some measurable property of the epigenome and/or transcriptome, and it is likely that a small subset of these could be positively selected. Traditional methods for inferring such selection would rely on finding recurrence of a particular genetic change across independent patient tumors, a well-trodden path for point mutations.[40–42] However, the challenge here is that chromothripsis accesses such a highly combinatorial space of potential genomic configurations that recurrence is exceedingly unlikely to occur. Instead, it may be that inference of positive selection in this setting will have to be done on the phenotypic effect rather than the genomic event, suggesting that studies such as this would need to be scaled to much larger sample sizes.

## Limitations of the study

High sequencing depth is required to generate haplotype-resolved genome assemblies. This is currently very costly and will be a limitation for using this technology for patient biopsies. Furthermore, mis-assemblies can be generated from subclonal events or incorrect haplotype resolution, and these are difficult to correct once the assembly is generated. The method currently generates two assemblies per chromosome, although in the future it could be adapted to produce more assemblies in cases with whole chromosome duplication or subclonal events.

## STAR★METHODS

Detailed methods are provided in the online version of this paper and include the following:

- KEY RESOURCES TABLE
- RESOURCE AVAILABILITY
  - Lead contact
  - Materials availability
  - Data and code availability
- METHOD DETAILS
  - Organoid culture
  - Genomic sequencing
  - Epigenetic sequencing
  - Sequence alignment
  - Genomic variant calling
  - De novo assembly methods
  - Assembly-based haplotype resolution
  - Structural variant calling
  - Peak calling in ChIP-seq and ATAC-seq data
  - Identifying differential transcript expressions
  - Identifying differential TAD structures

## SUPPLEMENTAL INFORMATION

## ACKNOWLEDGMENTS

This research was funded in whole, or in part, by the Wellcome Trust grant 206194. For the purpose of Open Access, the author has applied a CC BY public copyright license to any Author Accepted Manuscript version arising from this submission.

## AUTHOR CONTRIBUTIONS

J.I., P.J.C., and M.J.G. designed the study. J.I. cultured esophageal organoids, with guidance from H.E.F. M.A.Q. generated sequencing libraries for Hi-C. J.I. wrote and implemented a haplotype resolution algorithm. Z.N. generated the initial assemblies. E.H. scaffolded assemblies. Z.N. generated final assemblies by implementing CAUS. Z.N. generated dot plots. J.I. performed haplotype resolution of epigenetic and transcriptomic datasets and performed downstream analysis. J.I. generated variant calls. S.Z. and H.J. performed retrotransposon analysis. K.R. implemented integrated visualization of data. K.R., A.M., K.B., T.H.H.C., A.R.J.L., and D.L. contributed to formalization and evolutions of ideas for data analysis and algorithm development. J.I. and P.J.C. drafted the manuscript.

## DECLARATION OF INTERESTS

P.J.C. is an academic co-founder, stockholder, and consultant for Quotient Therapeutics.

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

## STAR★METHODS

### KEY RESOURCES TABLE

| REAGENT or RESOURCE | SOURCE | IDENTIFIER |
|---|---|---|
| **Antibodies** | | |
| CTCF | Diagenode | Cat# C15410210, RRID: AB_2753160 |
| H3K4me3 | Diagenode | Cat# C15410003, RRID: AB_2924768 |
| H3K27me3 | Diagenode | Cat# C15410195, RRID: AB_2753161 |
| H3K27ac | Diagenode | Cat# C15410196, RRID:AB_2637079 |
| **Deposited data** | | |
| Illlumina short read sequencing | This study | EGA dataset ID: EGAD00001010871, study ID: EGAS00001003264 |
| PacBio sequencing | This study | EGA dataset ID: EGAD00001010871, study ID: EGAS00001007163 |
| Hi-C | This study | EGA dataset ID: EGAD00001010871, study ID: EGAS00001003122 |
| 10x | This study | EGA dataset ID: EGAD00001010871, study ID: EGAS00001003191 |
| ChIP-seq | This study | EGA dataset ID: EGAD00001010871, study ID: EGAS00001007180 |
| ATAC-seq | This study | EGA dataset ID: EGAD00001010871, study ID: EGAS00001003890 |
| Iso Seq | This study | EGA dataset ID: EGAD00001010871, study ID: EGAS00001004051 |
| **Critical commercial assays** | | |
| Qiagen DNA Maxi Kit | Qiagen | Cat no: 13362 |
| Bionano Prep Blood and Cell Culture DNA Isolation Kit | Bionano | Cat no: RE-016-10 |
| Dovetail Genomics Hi-C library preparation kit | Dovetail | SKU: 21004 |
| iDeal ChIP-seq kit | Diagenode | Cat no: C01010055 |
| NEBNext Ultra II FS DNA Library Prep | New England Biolabs | Cat no: E6177 |
| Qiagen AllPrep DNA/RNA/miRNA Universal Kit | Qiagen | Cat. no.: 80224 |
| **Software and algorithms** | | |
| Gap5 | Bonfield et al.[22] | http://staden.sourceforge.net |
| 3D-DNA | Dudchenko et al.[27] | https://github.com/aidenlab/3d-dna |
| BLAT | Kent[43] | https://github.com/djhshih/blat |
| BRASS | Nik-Zainal et al.[44] | https://github.com/cancerit/BRASS |
| BWA | Li and Durbin[45] | https://github.com/lh3/bwa |
| CAUS | NA | https://github.com/wtsi-hpag/caus |
| CaVEMan | Jones et al.[46] | https://github.com/cancerit/CaVEMan |
| Cutadapt | Martin, M[47] | https://cutadapt.readthedocs.io/en/stable/installation.html |
| DeepVariant | oplin et al.[48] | https://github.com/google/deepvariant |
| DESeq2 | Love, Huber and Anders[35] | https://bioconductor.org/packages/release/bioc/html/DESeq2.html |
| DiffBind | Ross-Innes et al.,[34] | https://bioconductor.org/packages/release/bioc/html/DiffBind.html |
| GRIDSS | Cameron et al.[49,50] | https://github.com/PapenfussLab/gridss |
| hifiasm | Cheng et al.[26] | https://github.com/chhylp123/hifiasm |
| IsoSeq v3 | Zhang et al.[13,51] | https://github.com/macs3-project/MACS |
| MACS2 | Zhang et al.[13,51] | https://github.com/macs3-project/MACS |
| MEIGA-LR | Pacbio | https://github.com/PacificBiosciences/pbbioconda |
| NGMLR | Sedlazeck et al.[52] | https://github.com/philres/ngmlr |
| Pindel | Ye et al.[53] | https://github.com/genome/pindel |
| Skewer | Jiang et al.[54] | https://github.com/relipmoc/skewer |
| Sniffles | Sedlazeck et al.[52] | https://github.com/fritzsedlazeck/Sniffles |
| Strelka2 | Kim et al. | https://github.com/Illumina/strelka |

*(Continued on next page)*

*Continued*

| REAGENT or RESOURCE | SOURCE | IDENTIFIER |
|---|---|---|
| WhatsHap | Patterson et al.[25,55] | https://github.com/whatshap/whatshap |
| wtdbg2 | Ruan and Li[23] | https://github.com/ruanjue/wtdbg2 |
| ChromoDethripsis | This study | https://github.com/jannatjiaz/ChromoDethripsis |
| AssemblyBasedHaplotypeResolution | This study | https://github.com/jannatjiaz/AssemblyBasedHaplotypeResolution |

## RESOURCE AVAILABILITY

### Lead contact
Further information should be directed to and will be fulfilled by the lead contact, Peter J. Campbell (pc8@sanger.ac.uk).

### Materials availability
This study did not generate new unique reagents.

### Data and code availability
The datasets supporting the study have been deposited in the European Genome-Phenome Archive public repository: EGAD00001010871 (Table S5).

Scripts used for haplotype resolution of genomic datasets are publicly available and can be found at https://github.com/jannatjiaz/ChromoDethripsis (https://doi.org/10.5281/zenodo.10201498).

Scripts used for haplotype resolution of epigenomic datasets are publicly available and can be found at: https://github.com/jannatjiaz/AssemblyBasedHaplotypeResolution (https://doi.org/10.5281/zenodo.10201514).

Newly written scripts used to plot main figures are also publicly available and can be found here: https://github.com/jannatjiaz/ChromothripsisPaperFigures (https://doi.org/10.5281/zenodo.10202017).

## METHOD DETAILS

### Organoid culture
Five esophageal adenocarcinoma organoid cell lines, from four different patients, were derived by the Cellular Genotyping and Phenotyping team at the Wellcome Sanger Institute using a previously described method[56] (Table S6). HCM-SANG-0311-C15-B and HCM-SANG-0311-C15 were derived from the same patient. Forty five esophageal organoids were sequenced using Illumina short-read sequencing[57] and the 5 organoids used in this study were selected by calculating B-allele frequency (BAF) at sites of SNPs. This allowed identification of copy number states along a chromosome to query whether chromothripsis was restricted to only one allele.

To limit the number of culture-associated changes, cells were cultured for either a maximum of 30 weeks or below 30 passages from the point of banking, whichever occurred first. Organoids were passaged every 7–14 days, depending on the specific growth rate of the organoid.

### Genomic sequencing
Multiple sequencing platforms were used. For Illumina short-read sequencing, the Qiagen DNA Maxi Kit protocol (Qiagen, Germantown, MD USA) was used to extract DNA. Short insert 500 bp genomic libraries were constructed, flow cells were prepared and clusters were generated using standard Illumina PCR free library protocols (Illumina Inc, San Diego, USA).[58] The Illumina HiSeq2500 platform was used to generate 150 bp paired-end whole genome sequencing with 32-43x coverage. Matched normal blood cells were also sequenced with 32-88x using this methodology (Table S1).

For the 1M PacBio continuous long read (CLR) sequencing, high molecular weight DNA was extracted using the BioNano plug-based preparation protocol (BioNano, San Diego, CA USA). Libraries were sequenced on the PacBio Sequel. DNA for the 8M CLR and Circular Consensus Sequencing (CCS) PacBio sequencing was extracted using a Qiagen MagAttract extraction protocol and then sequenced on the PacBio Sequel. The CCS reads were size selected at 12 kb. Coverage for PacBio CCS and CLR reads ranged from 46 to 72x and 35-101x, respectively. Linked-reads were generated using 10X Chromium and coverage ranged from 30 to 35x.

Karyotypic data was also generated for all cell lines by the Karyotyping facility at the Wellcome Sanger Institute. Cells were arrested in metaphase using 100 ng/ml colcemid for 3 h and 20 metaphase cells were karyotyped per sample.

### Epigenetic sequencing
The Hi-C libraries were produced using the Dovetail Genomics Hi-C library preparation kit (Dovetail Genomics, Scotts Valley, CA USA) and the crosslinking for cell lines protocol. DpnII was used to fragment the DNA. Flow cells were prepared

and Illumina short-read libraries were sequenced on a HiSeq X Ten using recommended protocols. Total coverage generated was 115-121x.

For ChIP-seq, DNA was crosslinked using the Diagenode (Denville, NJ, US) cell fixation protocol. Formaldehyde and subsequently glycine were added directly to organoids in basement membrane extract, type 2 (BME-2). Organoids were then dissociated using TrypLE and stored in fixation buffer. Chromatin was prepared using the iDeal ChIP-seq kit for Transcription Factors protocol and im-munoprecitation was done using antibodies against H3K4me3, H3K27me3, H3K27ac and CTCF. Libraries were sequenced on the Illumina NovaSeq 6000 to generate 100 bp paired-end sequencing reads. Each mark had 7-9x coverage.

For ATAC-seq, the Fast-ATAC protocol was followed.[59] Cells were incubated with Tn5 transposase, which fragments DNA in open chromatin regions and adds adapters simultaneously. Cells were subsequently lysed using digitonin. Library preparation was done using qPCR-only Nextera Dual indexing, flow cells were prepared and DNA was sequenced on a HiSeq V4 to produce 75 bp paired-end sequencing reads. The coverage was 6-7x.

For Iso-seq, RNA was extracted using the Qiagen AllPrep DNA/RNA/miRNA Universal Kit (Qiagen, Germantown, MD USA). The extracted RNA had a RIN ≥7. Iso-seq libraries were constructed from the extracted RNA using the Iso-seq Express Template Prep-aration for Sequel and Sequel II Systems protocol. These libraries were sequenced by core facilities at the Wellcome Sanger Institute on the Sequel II with one SMRT cell per sample leading to 7-12x coverage.

### Sequence alignment

GRCh38 was used as the reference genome. The Illumina short-read, ATAC-seq reads, ChIP-seq reads and the Hi-C reads were aligned using the Burrows-Wheeler algorithm (BWA mem) (v0.7.17-r1188).[45] The ChIP-seq adapters were removed using cuta-dapt[45,47] and the ATAC-seq adapters were removed using Skewer.[54] The Hi-C reads were aligned as single-ended reads. The Illu-mina short-read, ATAC-seq reads and ChIP-seq reads were aligned as paired-end reads. The PacBio CCS, CLR and Iso-seq reads were aligned using NGMLR (v0.2.7),[52] after CCS consensus sequences were generated for the CCS and Iso-seq reads using ccs (v5.0.0). Primers and polyA tails in the Iso-seq reads were removed using lima (v2.0.1) and isoseq3 refine (v5.0.0), respectively, as part of the PacBio SMRTAnalysis software. Comparison of sequences to the reference was done using BLAT version 36x2.[43]

### Genomic variant calling

In the short reads from the organoids, single nucleotide variants (SNVs) were called using CaVEMan (v1.11.1)[46] and filtered on PASS variants, ASMD ≥140 and CLPM = 0. Indels were called using Pindel (v2.2.5).[53] SVs were called using BRASS (v6.2.0)[44] and vali-dated by the BRASS implementation of local assembly. In the Illumina short-read matched normal blood cell sequencing, germline variants were called using Strelka2 (Kim et al., 2018) and SVs were called using both BRASS (v6.2.0) and GRIDSS (v2.11.1).[49] In the PacBio CCS and CLR reads, SVs were called using Sniffles (v1.0.9).[52] SNVs and indels were called using DeepVariant.[48]

For *de novo* haplotype resolution, WhatsHap[55] was used to phase germline heterozygous variants and generate phase blocks on the chromosomes using only CCS reads. Heterozygous variants were called in matched normal blood and were filtered to remove regions of low complexity so only the most confident single nucleotide variant calls were used.

Due to the high density of SVs on the chromothriptic chromosome in HCM-SANG-0300-C15, an initial consensus set of haplotype-unresolved SVs were generated by Sniffles in the CCS and CLR reads. These SVs were validated in the raw CCS and raw CLR reads. SVs without support were removed, while those that were not called by Sniffles but were found and validated in the raw reads were rescued. These SVs were used to assign phase blocks to the wild-type and chromothriptic chromosomes, with the expectation that the chromothriptic chromosome would have more structural variants than the wild-type chromosome. The chromothriptic regions in the other samples had lower density of SVs than HCM-SANG-0300-C15 and therefore the structural variants were less informative for phasing. Instead, variant allele fractions (VAFs) of heterozygous single nucleotide polymorphisms (SNPs) and local read depths were generated for each phase block and were used to assign blocks to the appropriate allele. VAFs and read depth were also used to complement the SV-based phasing for HCM-SANG-0300-C15.

Together, structural variants, VAFs and regions of loss of heterozygosity (LOH) present on either allele allowed phase blocks with informative features to be confidently assigned. Where there were no informative variants, phase blocks were randomly assigned.

### De novo assembly methods

Hifiasm (v0.7)[26] was used to produce assemblies for all chromosomes other than chromosome 16 in HCM-SANG-0310-C15 where the hifiasm assembly did not accurately represent the underlying genomic sequence. In this case, Wtdbg2 (v3.3)[23] was used. Assem-blies were produced for each allele separately. The initial assemblies were then scaffolded using 3D-DNA (v180922).[27] The scaf-folding tool 3D-DNA makes many breakpoints when examining Hi-C read coverage profiles. As a result of this, scaffolds and contigs are sometimes excessively fragmented. To fix this feature, a modified version of CAUS was run (v1.0) (Chromosome Assembly using Synteny, https://github.com/wtsi-hpag/caus).

### Assembly-based haplotype resolution

Hi-C reads, ATAC-seq reads, ChIP-seq reads and Iso-seq reads were all haplotype resolved using alignment-based methods and the custom assembly generated for each chromosome. All reads were aligned to both the chromothriptic assembly and the wild-type assembly and assigned based on: presence of a heterozygous SNP, mapping into regions of LOH, mapping quality and mapping

distances of read pairs when appropriate. For Hi-C reads a weighted probability was used to determine which assembly the observed mapping distance was likely to be from. For Iso-seq reads, read alignments allow for splicing of transcripts. Regions without SNPs and SVs where sequences were identical in both assemblies would provide identical mapping scores and therefore when identical mapping was seen, reads were randomly assigned.

### Structural variant calling

RTs in the long read sequencing were called on each haplotype using MEIGA-LR v1.2.0 (manuscript in preparation) on haplotype-resolved CCS reads. Detected RT events were considered to be somatic when not described in a database of retrotransposon polymorphisms[60] and signs of RT were not detected in the following settings: Illumina data derived from matched normal blood, tumor CSS data of homologous chromosomes from the same donor and tumor CSS data from the other donors in the cohort. From this retrotransposition insertion events were called.

In the short reads, somatic RTs were called using the Genomic Rearrangement Identification Software Suite (GRIDSS) with default settings (version 2.9.4).[50] Retrotranspositions called in the matched normal were removed and somatic retrotranspositions supported by at least six reads (at least two split reads required) were included. Repeatmasker was used to annotate somatic RTs[61] and L1 RTs with QUAL $\geq$ 350 and Alu (and SVA) retrotranspositions with QUAL $\geq$ 700 were included.

SVs were called using haplotype-resolved PacBio CCS reads aligned to the GRCh38 reference genome. Since matched normal CCS reads were not available, germline SVs could not be filtered out based on a haplotype-resolved normal genome. Instead, germline homozygous SVs were filtered out if they were called on both alleles, identified in matched normal Illumina short reads, map to decoy or repeat regions or were retrotransposon events.

### Peak calling in ChIP-seq and ATAC-seq data

ATAC-seq and ChIP-seq peaks were called using MACS2.[51] Broad peak calling was used for H3K27me3 reads and narrow peak calling was used for all other ChIP-seq marks and for ATAC-seq reads. Differential binding was called using DiffBind.[34] To do this, reads were mapped back to GRCh38. Only reads which were explicitly assigned were included in this analysis.

### Identifying differential transcript expressions

Raw Iso-seq reads were processed using IsoSeq v3. DESeq2[35] was used to determine which transcripts were differentially expressed (Wald test, q value <0.05). Reads were mapped back to GRCh38 and only reads that were explicitly assigned were used.

### Identifying differential TAD structures

Hi-C contact matrices generated using Juicer[62] with default parameters. TAD boundaries were called using CoolTools.[36]

