## [Document S2. Transparent peer review records for Ijaz et al · Cell Genomics]

HAPLOTYPE-SPECIFIC ASSEMBLY OF SHATTERED CHROMOSOMES IN ESOPHAGEAL ADENOCARCINOMAS

Jannat Ijaz^{1,2*}, Edward Harry¹, Keiran Raine^{1,3}, Andrew Menzies¹, Kathryn Beal¹, Michael A. Quail¹, Sonia Zumalave⁴, Hyunchul Jung¹, Tim HH Coorens^{1,5}, Andrew RJ Lawson¹, Daniel Leongamornlert¹, Hayley E Francies^{1,6}, Mathew J Garnett¹, Zemin Ning¹, Peter J Campbell^{1,7,*}.

Summary

Initial submission: Received : 5/15/2023
Scientific editor: Judith Nicholson

First round of review: Number of reviewers: 2
Revision invited : 6/22/2023
Revision received : 10/13/2023

Second round of review: Number of reviewers: 2
Accepted : 12/11/2023

Data freely available: Yes

Code freely available: Yes

This transparent peer review record is not systematically proofread, type-set, or edited. Special characters, formatting, and equations may fail to render properly. Standard procedural text within the editor's letters has been deleted for the sake of brevity, but all official correspondence specific to the manuscript has been preserved.

Referees' reports, first round of review

Reviewer #1: In this manuscript, Jannat Ijaz et al. assembled oesophageal cancer genomes by performing short and long read sequencing and Hi-C analysis using organoids. They evaluated several methods to obtain cancer genome assemblies and found that haplotype-aware assemblies are useful to characterize their highly scattered chromosomes. Using the constructed haplotype-resolved genome assemblies, they also attempted to understand differences of transcriptional regulation and gene expression between chromothriptic and wild-type chromosomes. They found that parts of chromothriptic events would alter epigenomic and transcriptomic landscape. The comparison between chromothriptic and wild-type chromosomes in this study would be interesting for cancer genome researchers. There are some points that need to be addressed as below;

Points;

1. The authors mentioned that they obtained karyotypic data. However, the reviewer could not find the result in the manuscript. There are often problems about existence of aneuploidy and ecDNA in cancer genomes for such haplotype-aware analyses. The reviewer understand that the authors selected simple cases. However, the authors should show the data of karyotypes and discuss about these challenges.
2. The authors should show potential functions of the genes, such as AKAP12, TPD52L1, MYO6 and so on, in which transcriptional regulation was affected by chromothripsis, for carcinogenesis and acquirement of therapeutic resistance. Are they driver or passenger events? The authors also should discuss frequencies of occurrence of these events in oesophageal cancers.
3. Basic characterization of cancer genomes, such as OncoPrint for statuses of oncogenes/tumor-suppressor genes, Circos plot and so on, is still a useful information for understanding the biology of cancer in each organoid.

Reviewer #2: The manuscript by Ijaz et al describes their analytical pipeline for reconstructing shattered chromosome in five organoids and correlates transcriptomic and epigenetic profiles with the features of the rearranged chromosomes. This type of haplotype-specific analysis is of great interest to those interested in cancer genome analysis but is also very difficult. So, although the analysis described in the paper can be improved and biological insights obtained is limited despite many accompanying epigenetic data, I think the manuscript represents a strong effort for an important problem.

1. The abstract mentions subclonal variants as motivation for haplotype specific genomic analysis. But the research presented in this manuscript is work on organoids which were derived from cell lines with clonal chromothripsis. The described assembly does not indicate the handling of subclonal variants, and this should be made explicit in the abstract, possibly even in the title.
2. The authors selected samples with clonal chromothripsis affected single parental copies of the chromosome. Could the authors share their insights on the feasibility of performing similar analysis for real patient samples? I think it's fine to do what the authors did (organoids), but they should also state the limitations of their study.
3. Is there a reason that the authors did not use the Nanopore long reads? I think the ultra long reads would have been ideal for the current study. I am not asking that the analysis should be redone at this point; I'm just wondering if there is a reason not to do it if there is HiC data.
4. There is only one brief paragraph on retrotransposon analysis, so I wondered if the authors could expand a little more, even just on the technical side. How much is gained by haplotype-specific assembly compared to standard Illumine-only analysis? Is there anything we learn from haplotype-specific assembly that we didn't from non-haplotype analysis?

5. The chromatin association analysis is the least satisfying (Fig 4 and 5). The p-values are often not particularly meaningful because there are so many data points and so more description of the effect sizes would be informative. Regarding the point about the p-value, the text at one point says that the two samples from the same donor (initial and relapse) have "similar peak heights in both samples (Pearson correlation coefficient $r=0.69$, $p=1.74 \times 10^{-176}$; Figure 4A) suggesting that histone modifications and chromatin accessibility is stable." Here the small p-value indicates that they profiles are different, which is the opposite of stable. So the point is that more nuanced and detailed analysis would be helpful. Are some of the differential ChIP-seq peaks (p10, not clear which factors or histone modifications they're referring to) contiguous in the genome, for example? Any specific analysis of CTCF or enhancers?

6. Statistical analysis could be reviewed. For instance, Wald test, Fisher's Exact test, Wilcoxon test are mentioned at various points when similar analyses are being done. Again, p-values are not that meaningful, e.g., Fig 5H has different distribution but the p-value is 0.95 presumably because the means are similar even though the distributions are not. Is testing the equality of the mean meaningful here, especially on a log scale? More examination of specific cases as in Fig 6 may give more insights than just simple p-values.

7. Similar, the "Topological associated domains" section presents examples of affected genes in a single sample, but a more systematic exploration of the changes in the 3D genome after chromothripsis would be of interest. How does the chromatin conformation change in the other 4 organoids with chromothripsis? How do TAD boundaries change? Do they get stronger, weaker? Do TADs get bigger, smaller or do they not change much despite of the drastic restructuring of the chromosome? Do TADs stay rather stable within fragments or do they form novel contacts across breakpoints?

8. The stability of gene expression is described for highly expressed genes. However, Figure 5A and 5B both show an increased expression of previously not expressed genes in this patient after relapse. Can the authors comment on this observation?

Minor:

1. The version that I received had figures with very low resolution. Each figure had an embedded link for high resolution figures, but they were also of poor resolution. I had contacted the editorial office, but they were not able to help. I don't think this affected my evaluation of the paper materially but it did make it more difficult. The authors need to ensure that the figures are of high quality in the future.

2. For readability, the two paired samples should be next to each other (HCM-SANG-0311-C15-B and HCM-SANG-0311-C15) for comparison. For Fig 3 panels, why do some samples show genome averages and others only single chromosomes?

3. "HiSeq" or "Illumina" would be a better name than "X Ten"

4. I thought Supp 5A on the proportion of reads that could be assigned to a haplotype is an important figure. Perhaps that could be moved to a main figure. The description right now is simply that long reads are better than shorter reads. Some numbers could be informative.

Authors' response to the first round of review

Re: Haplotype-specific assembly of shattered chromosomes in oesophageal adenocarcinomas (CELL-GENOMICS-D-23-00136)

Thank you for the opportunity to provide a response to the comments from the reviewers of our paper. We are delighted that the reviewers found that the work addresses important questions that would be of interest to those in the cancer genomic field. Below, the reviewer comments are in blue, with our response in black and actions that we have undertaken for the revision in red and bold.

Reviewers' Comments:

Reviewer #1: In this manuscript, Jannat Ijaz et al. assembled oesophageal cancer genomes by performing short and long read sequencing and Hi-C analysis using organoids. They evaluated several methods to obtain cancer genome assemblies and found that haplotype-aware assemblies are useful to characterize their highly scattered chromosomes. Using the constructed haplotype-resolved genome assemblies, they also attempted to understand differences of transcriptional regulation and gene expression between chromothriptic and wild-type chromosomes. They found that parts of chromothriptic events would alter epigenomic and transcriptomic landscape. The comparison between chromothriptic and wild-type chromosomes in this study would be interesting for cancer genome researchers.

There are some points that need to be addressed as below;

We thank the reviewer for these kind comments.

1. The authors mentioned that they obtained karyotypic data. However, the reviewer could not find the result in the manuscript. There are often problems about existence of aneuploidy and ecDNA in cancer genomes for such haplotype-aware analyses. The reviewer understand that the authors selected simple cases. However, the authors should show the data of karyotypes and discuss about these challenges.

This is an excellent point raised by the reviewer. The assemblies represent the dominant clone of the maternal and paternal allele so our method is most suitable for cancer samples which contain only two haplotypes. However, in practice, most cancer samples will contain some degree of aneuploidy. From the karyotyping, it is evident that this is true for some of the chromosomes in our samples. **We have added in a supplementary figure containing the karyotyping data (Supplementary figure 2).** It is important to note that a lot of the subclonal variation occurs as a result of whole genome duplication events which should not affect the assembly as both parental alleles have been duplicated. However, whole chromosome duplications affecting only one haplotype will often lead to better haplotype resolution of reads. This is because haplotype blocks without informative SNPs can be assigned using read depth, reducing the number of randomly assigned block. Randomly assigned blocks should not meaningfully impact the assembly as they are identical between haplotypes, with no SNPs, no structural features or only copy number neutral structural features. However in practice, randomly assigned blocks lead to a more fragmented assembly.

An example of the increased contiguity of regions assembled from asymmetric compared to symmetric regions is evident when comparing HCM-SANG-0311-C15-B and HCM-SANG-0311-C15 (initial and relapse samples). Chromosome 9 is chromothriptic in both samples. In the initial sample there are two copies of the wild-type chromosome and one copy of the chromothriptic; however, in the relapse one of the wild-type alleles was lost. While the resultant assemblies are very similar, the asymmetric copy number in the initial sample led to a more contiguous assembly of the wild-type chromosome. The N90 in the initial sample was 40,086,745 for the wild-type chromosomes, with a L90 of 2. Comparatively the N90 in the relapse sample was 5,427,947, with a L90 of 5. The chromothriptic assemblies were more similar with an N90 of 3,358,574 and 5,819,878 for the initial and relapse, respectively, and an L90 of 5 and 4 for the initial and relapse, respectively. The increased read depth led to a more contiguous wild-type assembly when an asymmetric copy number is present. **We have included a more detailed explanation of how asymmetric copy numbers impact the assembly, including N90 and L90 stats for an example. We have also commented on the impact of whole genome duplication events which are frequently seen in our samples. This can be found in paragraph 5 in "Genome assemblies."**

As the reviewer notes, extrachromosomal DNA can also be difficult to assemble. It is difficult to detect in karyotyping data as the fragments are very small, although often they can be seen as small marker chromosomes or extrachromosomal fragments. However, the signatures of ecDNA can be readily detected in sequencing data from the large copy number jumps at the edges of the amplified segments (Campbell et al., 2008; Shoshani et al., 2021).

We did not detect any such jumps, and so we believe that these samples do not have substantial amounts of ecDNA. **We have included a paragraph that highlights that ecDNA is difficult to assemble and are unlikely to be included in our assemblies. This can be found in paragraph 3 in "Reconstructing other classes of SV."**

A) WTSI-OESO_103

B) WTSI-OESO_143

C) WTSI-OESO_148

D) WTSI-OESO_117

E) WTSI-OESO_152

2. The authors should show potential functions of the genes, such as AKAP12, TPD52L1, MYO6 and so on, in which transcriptional regulation was affected by chromothripsis, for carcinogenesis and acquirement of therapeutic resistance. Are they driver or passenger events? The authors also should discuss frequencies of occurrence of these events in oesophageal cancers.

To identify whether genes that are disrupted on chromothriptic chromosomes are driver or passenger events, we compared the differentially expressed genes between chromothriptic and wild-type chromosomes with drivers reported by the Oesophageal Cancer Clinical and Molecular Stratification (OCCAMS) Consortium (Frankell et al, 2019) and the ICGC/TCGA Pan-Cancer Analysis of Whole Genomes (PCAWG) Consortium. There is evidence of driver mutations on chromothriptic chromosomes that result from structural variation. This includes CDKN2A on chromosome 9 in HCM-SANG-0311-C15 which is lost by deletion on the chromothriptic chromosome and MAP3K4 in HCM-SANG-0300-C15, which is directly fragmented by an SV. The other samples had no evidence of driver mutations. We also ran gistic (Mermel et al., 2011) to identify recurrent copy number changes across the samples that are functionally important, including samples sequenced in TCGA. We found evidence of high copy number amplification: 3 in HCM-SANG-0300-C15, 4 on WTSI_OESO_117, 7 on WTSI_OESO_143, 14 on WTSI_OESO_148 and none on WTSI_OESO_152. Of those, very few were on chromothriptic chromosomes. HCM-SANG-0300-C15, HCM-SANG-0307-C15 and HCM-SANG-0311-C15 had none on the chromothriptic chromosome, HCM-SANG-0311-C15-B had 1 and HCM-SANG-0310-C15 had 2.

We have included the analysis of drivers and passenger events on chromothriptic chromosomes in all samples. This can be seen on paragraph 2 in "Allele-specific gene expression."

3. Basic characterization of cancer genomes, such as OncoPrint for statuses of oncogenes/tumor-suppressor genes, Circos plot and so on, is still a useful information for understanding the biology of cancer in each organoid.

This is an excellent suggestion. We have produced an oncoplot using drivers from the OCCAM dataset (Frankell et al, 2019) and circos plots for each organoid. This is now in supplementary figure 2.

Reviewer #2: The manuscript by Ijaz et al describes their analytical pipeline for reconstructing shattered chromosome in five organoids and correlates transcriptomic and epigenetic profiles with the features of the rearranged chromosomes. This type of haplotype-specific analysis is of great interest to those interested in cancer genome analysis but is also very difficult. So, although the analysis described in the paper can be improved and biological insights obtained is limited despite many accompanying epigenetic data, I think the manuscript represents a strong effort for an important problem.

We thank the reviewers for these kind comments.

1. The abstract mentions subclonal variants as motivation for haplotype specific genomic analysis. But the research presented in this manuscript is work on organoids which were derived from cell lines with clonal chromothripsis. The described assembly does not indicate the handling of subclonal variants, and this should be made explicit in the abstract, possibly even in the title.

Cancer genomes often contain subclonal variation, and this is observed in the karyotypic data (which we have now added as supplementary figure 3). These regions are difficult to

assemble as the read coverage in regions of subclonality are not equal to regions that are clonal. As such, most commonly these regions get collapsed by the assembler to resemble the most dominant clone.

Subclonal deletions that are present in only a few cells often do not have enough read coverage in order to reassemble that region. As such, short contigs are generated leading to a fragmented assembly in that region. An example of this is shown chromosome 13 in HCMSANG-

0300-C15. There is a subclonal deletion from ~70Mb to ~100Mb on haplotype 1 which is highly fragmented in the assembly. This is also evident in the karyotyping data where there are three different lengths of chromosome 13 in different cells. It is also worth noting that the subclonal loss of heterozygosity of this chromosomes makes assembling this region more difficult. In general, as the fraction of cancer cells containing the subclonal deletion decreases, the likelihood of assembling sequence from that region increases as there are more reads that will support the assembly.

Similar to clonal duplications, subclonal duplications are often collapsed by assemblers as most do not account for read depth. An example of this is shown in chromosome 15 in HCM-SANG-0300-C15. The karyotyping and short-read sequencing suggests there has been a subclonal whole chromosome duplication event of one allele which has subsequently undergone further rearrangement. In the genome assembly there is a normal haplotype and one haplotype with the initial deletion but not a second shorter haplotype. Unlike subclonal deletions, most subclonal duplications will be collapsed irrespective of how many cells containing the subclonal duplication.

We have included a supplementary figure showing the example outcomes as a result of subclonal events that are discussed above. We have also discussed how subclonal variants affect the genome assemblies in paragraph 3 in "reconstructing other classes of SVs."

Supplementary figure 6 - A) Rearrangement plots as previously described of chromosomes containing a subclonal deletion. B-C) Dot plots alignments of each haplotype to the reference GRCh38 genome. D) Karyotype for HCM-SANG-0311-C15 chromosome 13. E) Rearrangement plots as previously described of chromosomes containing a subclonal duplication. F-G) Dot plots alignments of each haplotype to the reference GRCh38 genome. H) Karyotype for HCM-SANG-0311-C15 chromosome 15.

2. The authors selected samples with clonal chromothripsis affected single parental copies of the chromosome. Could the authors share their insights on the feasibility of performing similar analysis for real patient samples? I think it's fine to do what the authors did (organoids), but they should also state the limitations of their study.

This is an interesting question. Our current implementation of these assemblies requires high sequencing depth and highly clonal samples as only the dominant clones are reconstructed. Organoids are a useful tool for this as they can be grown until there is sufficient amounts of DNA needed for the required sequencing. Our method to generate haplotype-resolved chromothriptic assemblies used circular consensus Pacbio, Hi-C and Illumina short-read, high-depth sequencing. We also needed epigenetic and transcriptomic sequencing to query differences between alleles.

Theoretically this analysis could be done from patient biopsies if the biopsy is large enough – we and others have successfully undertaken long-read sequencing and epigenetic profiling from primary tumour samples. Currently, however, assembly algorithms have mostly been designed for generating reference genomes and are ill-equipped to manage the complexities of cancer genomes, with their subclonality, contaminating normal cells and frequent regions of copy number gain. Nonetheless, we believe that this is an exciting area for further algorithmic development – one could build a patient-specific reference genome

from, say, a blood sample and then use this reference genome to deploy cancer-specific assembly methods that were aware of copy number gains and subclonality.

We have edited the manuscript to include this point to highlight that while technically possible to assemble chromosomes from patient samples, there are limitations that should be considered. This can be found in the first paragraph of the discussion.

3. Is there a reason that the authors did not use the Nanopore long reads? I think the ultra long reads would have been ideal for the current study. I am not asking that the analysis should be redone at this point; I'm just wondering if there is a reason not to do it if there is HiC data.

The reviewer is correct, these assemblies could have been generated using Nanopore sequencing. To get quality haplotype phased assemblies for cancer samples, PacBio HiFi or Oxford Nanopore duplex (Q20 or Q30) reads are needed. However, we were unable to get access to the Nanopore duplex data at the time of assembly production. When the sequencing reads were generated, the base-call error rate in Pacbio CCS reads was lower than the error rate in Nanopore reads. Since we assign reads to one haplotype block over the other based on SNPs, we worried that a higher error rate would lead to incorrect assignment of reads with more errors. Indeed, the longer length of ONT reads would help to increase the contig length. However we opted for shorter haplotype blocks that were highly accurate but generated a more fragmented assembly that could be later scaffolded. This was preferential to misassigning the ultra-long reads and then attempting to correct errors that may have arisen from that misassignment. However, both Nanopore and Pacbio reads could both be used as input for this pipeline. We have noted in the manuscript that it is possible to use Nanopore sequencing with our pipeline. This can also be found in the first paragraph of the discussion.

4. There is only one brief paragraph on retrotransposon analysis, so I wondered if the authors could expand a little more, even just on the technical side. How much is gained by haplotype-specific assembly compared to standard Illumine-only analysis? Is there anything we learn from haplotype-specific assembly that we didn't from non-haplotype analysis?

This is an excellent suggestion and we have now included data comparing short and longread sequencing for calling retrotransposition events. The majority of calls are present in both technologies (974). Of the unique calls, 472 were long-read specific and 201 and shortread specific. To understand why there were different calls in the long-read and short-read sequencing, we compared the GC content of the sequence around the retrotransposons and whether they were in repeat regions. While the GC content of the regions was, in general, similar between the calls specific to long-read and those specific to short-read sequencing, there was a set of long-read retrotranspositions missed by short-read sequencing with very low GC content. Furthermore, the long-read specific calls were also more frequent in repetitive regions. Taken together, this suggests that long-read sequencing is picking up a set of somatic retrotranspositions in regions with low sequence complexity where the longer read-length improves the confidence of mapping a de novo insertion of a repeat. We have included a comparison of long and short read retrotransposon calls in this manuscript. These figures can be found in Supplementary Figure 8 and the discussion of the different calls can be found in paragraph 1 of "Haplotype-resolved SV calls."

Number of overlapping retrotransposon calls when comparing calls from long-reads and short reads. GC content of 100 bp surrounding the retrotransposon insertion in all calls from the short-read sequencing compared to the long read sequencing specific calls. Fraction of retrotransposons that are in repeat versus non-repeat regions when comparing all shortread calls and long-read specific cells.

5. The chromatin association analysis is the least satisfying (Fig 4 and 5). The p-values are often not particularly meaningful because there are so many data points and so more description of the effect sizes would be informative. Regarding the point about the p-value, the text at one point says that the two samples from the same donor (initial and relapse) have "similar peak heights in both samples (Pearson correlation coefficient $r=0.69$, $p=1.74 \times 10^{-176}$; Figure 4A) suggesting that histone modifications and chromatin accessibility is stable." Here the small p-value indicates that they profiles are different, which is the opposite of stable. So the point is that more nuanced and detailed analysis would be helpful. Are some of the differential ChIP-seq peaks (p10, not clear which factors or histone modifications they're referring to) contiguous in the genome, for example? Any specific analysis of CTCF or enhancers?

We thank the reviewer for this point. With regards the comparison between the initial and relapse samples, we should clarify that the p-value refers to the Pearson correlation coefficient – thus, the significance here denotes that the chromatin peaks on the chromothriptic and wild-type chromosome in the initial and relapse samples are highly correlated.

We agree with the reviewer's point that the p-values do not adequately communicate the scale of the correlation particularly well. Looking at the raw numbers when comparing the initial and relapse samples, there are 610 peaks on chromosome 6 and 519 peaks on chromosome 9 that are present in both the initial and the relapse organoids that we can accurately haplotype resolve. On a per mark level, we identified how many marks were present in both samples at the haplotype level. The majority of histone modifications and CTCF sites are present in both samples. It is important to note that the total peaks (plotted in Figure 4A-B) can be made up of one or more individual histone marks or CTCF binding sites. Therefore, when determining the stability of individual marks, the sum of individual marks is greater than the total number of peaks called.

To contextualise the stability over time, we have included a breakdown of the total

number of peaks considered and how many contained active/repressive or CTCF binding sites. This breakdown can be found in the first paragraph of "Allele specific chromatin accessibility and histone modifications" and venn diagrams illustrating the overlap can be found in Supplementary Figure 9. We have also included a breakdown of the total number of genes included in the equivalent analysis for the stability of gene expression, found in the first paragraph of "Allele specific gene expression."

Supplementary Figure 9: Overlap of histone modification and CTCF binding peaks on haplotypes in the initial and relapse samples. Light green: unique to HCM-SANG-0311-C15-B, blue: unique to HCM-SANG-0311-C15, dark green: shared. Most marks have a high overlap between samples.

6. Statistical analysis could be reviewed. For instance, Wald test, Fisher's Exact test, Wilcoxon test are mentioned at various points when similar analyses are being done. Again, p-values are not that meaningful, e.g., Fig 5H has different distribution but the p-value is 0.95 presumably because the means are similar even though the distributions are not. Is testing the equality of the mean meaningful here, especially on a log scale? More examination of specific cases as in Fig 6 may give more insights than just simple p-values.

This is an excellent point. We have given 4 examples of specific cases (Figure 6 and

Supplementary figures 11-13) where there have been differences between chromosomes, however it is difficult to get a sense of how generalisable these features are from specific examples. The reviewer is correct that associating an effect size with the statistical analysis would be useful for this. **For cases where we have used a Fisher's Exact test, we have added odds ratios and confidence intervals. For cases where we have used Wilcoxon test, median distances for distance plots and % of peaks that are within 10Kb of SVs have been reported.**

7. Similar, the "Topological associated domains" section presents examples of affected genes in a single sample, but a more systematic exploration of the changes in the 3D genome after chromothripsis would be of interest. How does the chromatin conformation change in the other 4 organoids with chromothripsis? How do TAD boundaries change? Do they get stronger, weaker? Do TADs get bigger, smaller or do they not change much despite of the drastic restructuring of the chromosome? Do TADs stay rather stable within fragments or do they form novel contacts across breakpoints?

We thank the reviewer for an interesting point. Some examples of altered TADs are in figure 6 and supplemental figures 11-13. TAD calling is difficult to do for complex cancer genomes. We hypothesize this is due to subclonal rearrangements that may alter TAD structures in only a fraction of cells. This will disproportionately affect the chromothriptic chromosome when compared to the wild-type chromosome as our haplotype resolution method is more likely to assign these reads to the chromothriptic chromosome. Furthermore, given the read length of Hi-C reads is 150 bp, only ~40 % of reads can be actively assigned, the rest are randomly assigned. Both problems cause TAD boundaries to become less clean and therefore lead to some TADs not being called.

Notwithstanding these caveats, we did undertake identification of TAD boundaries, as suggested, and this yielded some fascinating data. TAD boundary calling can be performed at different resolutions to detect coarse- or fine-grained chromosome organisation, so we used scales of 150kb, 250kb, 500kb, 750kb and 1Mb for TAD boundary calls. The distance between boundaries can then be used to infer TAD size. There was no significant difference (p-value=0.87, median WT size=330kb, median CT size=335kb) between the chromothriptic and wild-type in chromosome 6 in HCM-SANG- 0300-C15 when looking at TADs called at the 150kb resolution. However for TADs that can be called at 250kb and greater, TADs were, on average, significantly larger on the chromothriptic chromosome than the wild-type chromosomes (250kb: p-value=0.044, median WT size=440kb, median CT size=485kb. 500kb: p-value=0.018, median WT size=595kb, median CT size=730kb. 750kb: pvalue= 0.0035, median WT size=700kb, median CT size=868kb. 1Mb: p-value=0.0019, median WT size=705kb, median CT size=890kb). This suggests that after chromothripsis, the larger scale chromosome organisation is disrupted with more contacts between distant regions of the genome and less strict compartmentalisation of different segments into TADs.

We have also done the equivalent analysis on the chromothriptic chromosomes in other samples, however there was no significant difference between the TADs on wild-type versus chromothriptic chromosomes at any resolution. The most plausible explanation for this is that the lower density of rearrangements on these other chromosomes means that chromosome organisation is less disrupted by chromothripsis. **We added in a new main figure (now Figure 6) with effect sizes stated and a supplementary figure (Supplementary figure 13) and expanded on this analysis in the first paragraph of "Topological associated domains."**

A) Wild-type chromosome

B) Chromothriptic chromosome

C) TAD boundaries detected

Figure 6: A-B) Example TAD boundary calls on wild-type and chromothriptic chromosomes, respectively. Top: Hi-C contact matrix with SV track showing where structural variants are present relative to the reference genome. Middle: TAD boundary calls using 150kb bins. Signal values represent differences between adjacent bins. When the differences are considered to represent TAD boundaries, orange dots are plotted. These correspond with small TAD structures. Bottom: TAD boundary calls using 1Mb bins as described above. These represent larger TAD structures. C) TAD sizes on wild-type versus chromothriptic chromosomes in HCM-SANG-0300-C15 called using different bin sizes. TADs are inferred as regions between boundary calls.

Supplementary figure 13) TAD sizes on wild-type versus chromothriptic chromosomes in other samples with chromothripsis (excluding HCM-SANG-0300-C15) called using different bin sizes. TADs are inferred as regions between boundary calls.

8. The stability of gene expression is described for highly expressed genes. However, Figure 5A and 5B both show an increased expression of previously not expressed genes in this patient after relapse. Can the authors comment on this observation?

This is an interesting point regarding novel genes that are different between the initial and relapse sample. While most genes have similar expression in the initial and relapse samples from the same patient, there are some genes that are not expressed in one sample but become highly expressed in the other sample. Between initial sample, chemotherapy and relapse there is evolution of the cancer. This is evident by the presence of structural variants that are unique to the relapse sample. It is important to note that the altered gene expression may also derive from altered regulatory landscape that occurs independently of genomic alteration. There are 224 genes that have no expression on chromosome 6 in the initial sample but become expressed in the relapse sample. There are also 117 that are expressed on chromosome 6 in the initial sample but do not have any expression in the relapse sample. For chromosome 9, there were 133 genes that become expressed in the relapse sample but have no measurable expression in the initial sample. There are also 68 genes that are not expressed in the relapse sample but are expressed in the initial sample. Of these genes only 1 on chromosome 6 (CCND3) and 1 on chromosome 9 (SPTAN1) are reported as oesophageal cancer drivers.

We have reported the number of genes that have no expression in one sample but are expressed in the other. We have also reported how many of them are likely to be cancer drivers as reported by TCGA or OCCAMS. This can be found in paragraph 1 in "Allelespecific gene expression."

Minor:

1. The version that I received had figures with very low resolution. Each figure had an embedded link for high resolution figures, but they were also of poor resolution. I had contacted the editorial office, but they were not able to help. I don't think this affected my evaluation of the paper materially but it did make it more difficult. The authors need to ensure that the figures are of high quality in the future.

We thank the reviews for pointing this out, we will ensure that high resolution figures will be included in the final version.

2. For readability, the two paired samples should be next to each other (HCM-SANG-0311-C15-B and HCM-SANG-0311-C15) for comparison. For Fig 3 panels, why do some samples show genome averages and others only single chromosomes?

We thank the reviewer for this comment, since the rate of retrotransposition did not seem to be perturbed by chromothripsis we have shown the genome averages. However for structural variants, the rate to SV will be wildly different between the samples with and without chromothripsis so we have focused on the chromothriptic chromosome. Specific chromosomes have been mentioned in the axis if there are multiple chromosomes containing chromothripsis in a sample (HCM-SANG-0311-C15). But the reviewer is correct, this is confusing. **We have edited figure 3 axis labels. When the entire genome is being considered (e.g. retrotransposon events), we only use the sample name. When we are considering only the chromothriptic chromosomes we specify which chromosome it is even if there is only one chromothriptic chromosome in that sample. We have also added in the size distribution of SVs on other chromosomes in 3J and 3I for consistency with the rest of the figure.**

3. "HiSeq" or "Illumina" would be a better name than "X Ten"

We thank the review for this comment. We have used Illumina short-read sequencing

instead of X Ten .

4. I thought Supp 5A on the proportion of reads that could be assigned to a haplotype is an important figure. Perhaps that could be moved to a main figure. The description right now is simply that long reads are better than shorter reads. Some numbers could be informative.

We thank the reviewer for this comment. **We have moved this plot into a main Figure 3**

References:

Campbell, P.J. et al. (2008) 'Identification of somatically acquired rearrangements in cancer using genome-wide massively parallel paired-end sequencing', *Nature genetics*, 40(6), pp. 722–729.

Shoshani, O. et al. (2021) 'Chromothripsis drives the evolution of gene amplification in cancer', *Nature*, 591(7848), pp. 137–141.

Mermel, C.H. et al. (2011) 'GISTIC2.0 facilitates sensitive and confident localization of the targets of focal somatic copy-number alteration in human cancers', *Genome biology*, 12(4), p. R41.

Referees' reports, second round of review

Reviewer #1: The authors have satisfactorily addressed the reviewer's concerns.

Reviewer #2: The authors have addressed my comments satisfactorily.

Authors' response to the second round of review

None